# How Patterns Dictate Learnability in Sequential Data

Mario Morawski*     Anaïs Després*

Rémi Rehm

## Abstract

Sequential data—ranging from financial time series to natural language—has driven the growing adoption of autoregressive models. However, these algorithms rely on the presence of underlying patterns in the data, and their identification often depends heavily on human expertise. Misinterpreting these patterns can lead to model misspecification, resulting in increased generalization error and degraded performance. The recently proposed `evolving pattern (EvoRate)` metric addresses this by using the mutual information between the next data point and its past to guide regression order estimation and feature selection. Building on this idea, we introduce a general framework based on predictive information—the mutual information between the past and the future, $\mathbf{I}(X_{\text{past}}; X_{\text{future}})$. This quantity naturally defines an information-theoretic learning curve, which quantifies the amount of predictive information available as the observation window grows. Using this formalism, we show that the presence or absence of temporal patterns fundamentally constrains the learnability of sequential models: even an optimal predictor cannot outperform the intrinsic information limit imposed by the data. We validate our framework through experiments on synthetic data, demonstrating its ability to assess model adequacy, quantify the inherent complexity of a dataset, and reveal interpretable structure in sequential data.

## 1 Introduction

From time series in finance and healthcare [3, 45, 34] to text streams in natural language processing [38], much of the real-world data ingested by machine learning systems is inherently sequential. Autoregressive models are commonly trained to predict such data, with their performance relying heavily on capturing evolving patterns. However, identifying these patterns still depends largely on human expertise [55]. Misinterpretation can result in models that overfit to training data, leading to poor calibration, generalization errors, and heightened vulnerability to adversarial examples [22]; for instance, classifiers can memorize random label assignments in the training set [56]. A substantial body of work is devoted to bounding generalization error, primarily through Bayesian theory [2]. For sequential data, these bounds have been formulated using Rademacher complexity [36], leveraging concentration inequalities and Bayesian-inspired methods. We identify a critical open question in the study of sequential data: **(i) What is the minimal achievable risk for a predictor attempting to model sequential data?** This question differs subtly but importantly from bounding the gap between empirical and true risk. Empirical evidence, such as the plateau in model performance on the *Exchange* dataset from `GluonTs` [1] despite recent innovations [46], suggests that data limitations—rather than model inadequacy—may be the binding constraint. This leads to a critical second question: **(ii) Can we distinguish whether poor performance stems from the model's limitations or from the inherent unpredictability of the data?** This distinction hinges on the nature and strength of temporal patterns: fewer patterns imply that even an optimal predictor will perform poorly, while richer structure offers more room for effective forecasting. Thus, good

---

*Equal Contribution

39th Conference on Neural Information Processing Systems (NeurIPS 2025).

prediction requires both (1) that the data contain exploitable patterns, and (2) that the model can identify and leverage them. To illustrate this, consider the following stylized scenarios in which a colleague attempts to predict what meal a researcher will bring to lunch each day:

- Meals are sampled uniformly at random from a cookbook.
- Meals are chosen weekly based on preferences, weather, and prior meals.
- Meals follow a fixed rotation established years ago.

In the first case, even the optimal predictor cannot outperform random guessing due to the absence of structure. In contrast, the second and third cases involve latent patterns—periodicity, preference dependencies, contextual triggers—that a well-designed model could exploit. Crucially, identifying whether such structure exists—and how much of it is learnable—is a prerequisite for developing effective models. Concurrently, recent work has pointed out the lack of effective metrics for quantifying evolving patterns in sequential data, proposing the `evolving rate (EvoRate)` metric as a preliminary solution [55]. Building on this idea, we generalize the notion of `EvoRate` through the concept of *predictive information*—the mutual information between the past and the future [7, 14]. This leads to the formulation of the **universal learning curve** [7], which measures the rate at which predictive information increases with observation length. This curve serves as a fundamental tool to quantify the minimal achievable risk. Our framework thus provides an information-theoretic perspective that addresses both questions **(i)** and **(ii)**, while also offering insights into the nature and strength of the patterns embedded in the data. Our main contributions are:

- **Establishing** a theoretical link between the presence of temporal patterns in sequential data and the minimal achievable prediction risk.
- **Deriving** information-theoretic bounds on this minimal risk, expressed in terms of structural properties of the data such as predictive information and pattern complexity.
- **Proposing** a practical estimator of the intrinsic risk limit, enabling quantitative comparison with model performance to assess whether learning is limited by data or by the model.
- **Validating** the framework on synthetic data, showing how it supports model selection and reveals when further improvements are constrained by the data itself. The codes are available on GitHub: `https://github.com/EkMeasurable/Learnability_Ipred`

## 2 Related work

**Patterns Estimation for Sequential Data**  Early work on measuring predictability in time series includes `ForeCA` [23], which introduced the concept of forecastability, based on the entropy of the spectral density. This metric quantifies how predictable a time series is from a frequency-domain perspective. More recently, `EvoRate` [55] proposed a mutual information-based approach to capture evolving patterns in sequential data, and demonstrated its use for guiding regression order and feature selection. While promising, both `ForeCA` and `EvoRate` focus primarily on intrinsic signal properties and lack a principled connection to model performance. In particular, `EvoRate` emphasizes temporal changes in its value rather than its absolute magnitude, which remains under-exploited. Moreover, these methods do not assess whether a predictive model has effectively captured the patterns present in the data. In contrast, we introduce an information-theoretic estimator of the minimal achievable risk on a dataset, which serves two key purposes: (i) indicating the presence and strength of temporal patterns—lower risk suggests stronger patterns—and (ii) enabling direct comparison with a model's empirical risk.

**Learning under General Stochastic Processes.**  Recent advances in statistical learning theory have increasingly focused on relaxing the classical assumption of independent and identically distributed (i.i.d.) data. In particular, extending learning frameworks to handle stochastic processes with temporal dependencies or evolving distributions has become a central research direction. The *Prospective Learning* framework [49] formalizes this setting by requiring a learner to produce a sequence of hypotheses that achieve low risk on future observations, given the data observed up to the present—an approach particularly relevant in non-stationary or dynamic environments where the optimal predictor may change over time. Complementary efforts have characterized learnability under general stochastic processes [18, 25], providing general conditions for when consistent learning

is possible, often framed in terms of regret minimization or universal consistency. These perspectives share the common goal of determining whether, and under what conditions, learning remains feasible in complex non-i.i.d. settings. Our work complements these approaches by introducing an intrinsic, information-theoretic notion of learnability. Rather than relying solely on external performance measures such as risk or regret, we quantify learnability through the predictive structure of the data itself—captured by predictive information and the universal learning curve. This allows us to bridge theoretical learnability guarantees with the structural properties of the data-generating process, offering a complementary perspective on when and why generalization is possible in sequential environments.

**Mutual Information Estimation (MI)**  Traditional methods, such as the k-nearest neighbor estimator [30], perform well in low-dimensional settings but struggle when applied to long or high-dimensional time series. MINE [6] addresses these challenges by leveraging the Donsker–Varadhan representation and deep neural networks to learn flexible estimators of mutual information. However, MINE exhibits high variance and instability, particularly in sequential data settings. Alternative approaches, such as InfoNCE [40], originally developed for contrastive predictive coding (CPC), are better suited to time series tasks; they learn representations that maximize mutual information between past and future segments. Further advancements include CLUB [11], which provides a tractable upper bound on mutual information, and SMILE [51], which improves estimator stability through Jensen–Shannon–based objectives.

**Predictive Information and Learning Curve.**  Originally introduced by [7, 8], predictive information can be viewed as a generalization of `EvoRate`. The core idea is to measure the mutual information between a context, $X_{\text{past}}$, and a target, $X_{\text{future}}$. It has been applied in machine learning to learn meaningful data representations [33], and several methods for estimating it have been proposed [22]. Building on this line of work, [7] and [15] also introduced the notion of the *universal learning curve*, which describes how predictive information grows with the length of the observed context. This curve captures the fundamental limits of learnability by quantifying how much additional information about the future can be extracted from longer histories. In this paper, we extend these theoretical ideas by establishing a novel link between the universal learning curve and the *minimal achievable risk*. Specifically, we show that the learning curve can be used to derive an empirical estimator of the optimal prediction performance attainable by any model. This connection provides an operational interpretation of predictive information as a diagnostic tool to distinguish between model limitations and the intrinsic unpredictability of the data.

**Lowest Possible Error Rate**  In classification tasks, the Bayes error rate represents the lowest achievable error rate for any classifier [53, 10, 54]. This concept was developed alongside the PAC-Bayes framework, which provides upper bounds on the risk [47, 32]. In the context of sequential data, prior work has focused primarily on bounding the gap between empirical and true risk, notably through Rademacher complexity [37, 36, 31]. However, to the best of our knowledge, none of these approaches explicitly connects the minimal achievable risk to the presence of patterns in sequential data.

## 3    Preliminary

### 3.1    Notations and hypothesis

We consider a stochastic process $\mathbf{X}_t^T = \{X_u\}_{u=t}^T \in \mathcal{X}^{T-t+1}$, indexed over $[t, T]$, with $t \in [0, T]$. We assume:

$$(\mathbf{H_0}) : \quad \text{The process is stationary and } H(X_t) < \infty,$$

where $H(\cdot)$ denotes entropy. We do not require $\mathbb{E}(X_t^2) < \infty$, allowing for heavy-tailed distributions, and typically set $\mathcal{X} = \mathbb{R}^d$.

Given a predictor $g : \mathcal{X}^k \to \mathcal{X}$ and a loss function $\ell : \mathcal{X} \times \mathcal{X} \to \mathbb{R}$, we define the **forecasting risk** of order $k$ as:

$$\mathcal{R}^{(k)}(g) = \mathbb{E}_{\mathbf{X}_{t-k+1}^{t+1}} \left[ \ell(X_{t+1}, g(\mathbf{X}_{t-k+1}^t)) \, \big| \, \mathbf{X}_{t-k+1}^t \right]. \tag{1}$$

## 3.2 Entropy and Entropy Rate

Understanding the nature of sequential data is closely linked to studying the memory properties of the underlying process. For a stationary process $\mathbf{X}_t^T$, a fundamental quantity for measuring the information content is its entropy [48], defined as

$$H(\mathbf{X}_t^T) = -\int_S p(\mathbf{X}_t^T) \ln p(\mathbf{X}_t^T)\, d\mathbf{X}, \tag{2}$$

where $S$ denotes the support of the distribution. Under Assumption $(\mathbf{H_0})$, the joint density $p(x_t, \ldots, x_T)$ is invariant under time translation, so the entropy depends only on the length of the block $[t, T]$. The entropy is concave and subadditive with respect to the block size, and satisfies $H(0) = 0$. Let $l(k) = H(X_t \mid \mathbf{X}_{t-k+1}^{t-1}) = -\mathbb{E}_{P(X_t, \mathbf{X}_{t-k+1}^{t-1})} \ln P(X_t \mid \mathbf{X}_{t-k+1}^{t-1})$ be the **entropy rate** of order $k$ [13]. It measures how well $X_t$ can be predicted by observing $k$ past values. Naturally, the more past observations are available, the lower the uncertainty, so the mapping $k \mapsto l(k)$ is non-increasing and non-negative. Under Assumption $(\mathbf{H_0})$, $l(k)$ converges to a constant $l_0$ as $k \to \infty$, and the **fundamental theorem of entropy** [48] states that

$$\frac{1}{k} H(\mathbf{X}_{t-k+1}^t) \underset{k \to \infty}{\longrightarrow} l_0, \tag{3}$$

that is $H(k) \sim k l_0$ when $k$ is large.

## 3.3 $\mathbf{I}_{\text{pred}}$ as a generalization of `EvoRate`

In Zeng et al. [55], `EvoRate` is proposed as a measure for capturing predictive patterns in sequential data. It is defined as the mutual information between the past $k$ observations, $\mathbf{X}_{t-k+1}^t$, and the next observation, $X_{t+1}$, that we note $\mathbf{I}(\mathbf{X}_{t-k+1}^t; X_{t+1})$. We naturally extend `EvoRate` by introducing a fundamental quantity,

$$\mathbf{I}_{\text{pred}}(k, k') = \mathbf{I}(\mathbf{X}_{t-k+1}^t; \mathbf{X}_{t+1}^{t+k'}) = \int p(\mathbf{X}_{t-k+1}^{t+k'}) \ln \frac{p(\mathbf{X}_{t-k+1}^{t+k'})}{p(\mathbf{X}_{t-k+1}^t) p(\mathbf{X}_{t+1}^{t+k'})}\, d\mathbf{X}. \tag{4}$$

Predictive information, denoted $\mathbf{I}_{\text{pred}}$, quantifies how much information from the past can be used to predict the future. Under assumption $(\mathbf{H_0})$, it converges in the limit as both context lengths grow:

$$\lim_{k, k' \to \infty} \mathbf{I}_{\text{pred}}(k, k') = \mathbf{I}_{\text{pred}}(X_{\text{past}}, X_{\text{future}}), \tag{5}$$

a quantity also known in the literature as *excess entropy* [16, 14, 21] or the *effective measure of complexity* [24]. Our focus remains on finite $k$ and $k'$ values to ensure practical relevance. Under $(\mathbf{H_0})$ and suitable regularity conditions, predictive information corresponds to the sub-extensive component of the entropy.

The asymptotic behavior of predictive information provides a principled lens through which to assess the complexity and predictability of stochastic processes [7, 15]. In machine learning, recent works have leveraged this quantity to improve learned representations [33, 22], taking advantage of its ability to capture shared temporal structure across sequences.

## 4 Theoretical analysis of $\mathbf{I}_{\text{pred}}$

Consider the set $\mathcal{F}_k = \left\{ (\mathbf{X}_0^t, X_{t+1}) \mapsto -\ln Q\left(X_{t+1} \mid \mathbf{X}_{t-k+1}^t\right) \mid Q \in \mathcal{H}_k \right\}$, which defines a class of real-valued functions on $\mathbf{X}$. This class encompasses all possible loss functions that can be derived from making predictions using a model $Q \in \mathcal{H}_k = \{Q_\theta \mid \theta \in \Theta\}$, where $\mathcal{H}_k$ represents a family of models of order $k$ parameterized by $\theta$—the vector of model parameters that governs the behavior and structure of the prediction model.

### 4.1 From a learning perspective

The $k^{\text{th}}$-order forecasting risk $\mathcal{R}^k(Q)$, defined as

$$\mathcal{L}_{\text{mle}}^k = -\mathbb{E}_{P(X_{t+1}, \mathbf{X}_{t-k+1}^t)} \ln Q(X_{t+1} \mid \mathbf{X}_{t-k+1}^t), \tag{6}$$

has been widely used to assess autoregressive model performance [44, 55]. In a similar spirit, we draw a connection between the predictive information $\mathbf{I}_{\text{pred}}$ and a central concept in information theory: the *universal learning curve* $\Lambda(k) = \ell(k) - \ell_0$ [7, 15], also known as the *entropy gain*. This quantity measures the reduction in uncertainty about the future obtained by conditioning on $k$ past observations, and thus captures the presence of temporal patterns in sequential data. A key theoretical connection between the predictive information and the learning curve is established by the following result from Bialek and Tishby [7]:

**Proposition 4.1** (*Bialek and Tishby (1999)* [7]). *Under hypothesis* ($\mathbf{H_0}$), *we have:*

$$\mathbf{I}_{pred}(k+1, k') - \mathbf{I}_{pred}(k, k') \longrightarrow \Lambda(k) \quad as \quad k' \to \infty.$$

*Proof in Appendix A.6.*

The notion of a learning curve is central to our analysis. In `EvoRate` [55], the authors primarily focused on how the metric evolves with the size of the past window, as its absolute values were difficult to interpret. In contrast, the universal learning curve offers a more principled alternative: it can be seen as a discrete derivative of the predictive information [7], capturing the marginal contribution of each additional past observation. This perspective not only enhances interpretability but also plays a key role in our theoretical developments and empirical evaluations.

## 4.2 Interpreting the Asymptotic Behavior of $\Lambda(k)$

The asymptotic behavior of $\Lambda(k)$ offers insight into the nature of the temporal patterns present in the data. It helps characterize the structure of the underlying distribution and supports the theoretical link between $\Lambda(k)$ and the presence of predictive patterns. We now present examples that illustrate the relevance of this perspective.

**The special case of Markov processes.** When $X$ is generated by a Markov process of order $m$, dependencies are limited to the past $m$ observations. This structure is naturally reflected in the predictive information $\mathbf{I}_{\text{pred}}$. Previous approaches have addressed Markov order estimation using conditional mutual information [42], with `EvoRate` showing promising empirical results [55], though without formal guarantees. Classical criteria like AIC [29] and BIC [17] also remain standard tools.

Building on Crutchfield and Feldman [15], we derive a closed-form expression for $\mathbf{I}_{\text{pred}}$ under the Markov assumption and show that the universal learning curve $\Lambda(k)$ vanishes for all $k \geq m$, providing a theoretical link between pattern complexity and memory length.

**Proposition 4.2** (Predictive information in Markov processes). *Let $\mathbf{X}_t^T$ be a Markov process of order $m$. If $k' \geq k \geq m$, then:*

$$(i) \quad \mathbf{I}_{pred}(k, k') = \mathbb{E}_{\mathbf{X}_{t-m+1}^{t+m}} \left[ \ln \frac{P(X_{t+1}^{t+m} \mid \mathbf{X}_{t-m+1}^t)}{P(X_{t+1}^{t+m})} \right],$$

$$(ii) \quad \forall k \geq m, \quad \Lambda(k) = 0.$$

*In particular, for first-order Markov processes, we recover that $\mathbf{I}_{pred}(k, k') =$ `EvoRate`$(1)$ for all $k \geq 1$, allowing $\Lambda(k)$ to identify the true Markov order.*
*Proof in Appendices A.7 and A.8.*

**Parametric Processes.** Let the process $\mathbf{X}_{t-k+1}^t$ be generated by a parametric family $Q_{\mathbf{X}_{t-k+1}^t}(\theta)$, with unknown parameter $\bar{\theta}$ drawn from prior $\mathcal{P}(\theta)$ and $\dim \Theta = p$. Classical results in the i.i.d. case [7, 12] show that predictive information grows logarithmically with $k$. We extend this to dependent settings under mild regularity conditions.

**Theorem 4.3** (Predictive information in parametric models). *Assuming standard hypotheses on stationarity, weak dependence, and regularity of the parametric family, (1, 2, and 3), let the past and future windows grow with $k \to \infty$ and $k' \geq k$ (the ratio $k'/k$ may vary). Then*

$$\mathbf{I}_{pred}(k, k') \underset{k \to \infty}{=} \frac{p}{2} \ln(k) + \frac{1}{2} \ln \det(F) + \mathcal{O}(1),$$

*where $F$ is the Fisher information matrix associated with the divergence $D_{KL}(Q_{\mathbf{X}_{t-k+1}^t}(\theta) \,\|\, Q_{\mathbf{X}_{t-k+1}^t}(\bar{\theta}))$.*
*Proof in Appendix A.9.*

This result provides an interpretable asymptotic expansion: the first term captures the uncertainty due to parameter estimation, while the second reflects model confidence via the concentration of likelihood around $\bar\theta$. In contrast to EvoRate, whose asymptotic behavior remains unclear, $\mathbf{I}_{\text{pred}}$ exhibits a principled structure grounded in classical learning theory.

**Corollary 4.4** (Universal Learning Curve Decay). *Under the same assumptions, the universal learning curve satisfies:*

$$\Lambda(k) \sim \frac{p}{2k}.$$

*In particular, the dimensionality of the parameter space can be estimated as $\dim \Theta \approx 2k \, \Lambda(k)$.*
*Proof in Appendix A.9.*

This relation reveals a precise connection between the decay of the universal learning curve $\Lambda(k)$ and the intrinsic dimensionality $p$ of the parameter space $\Theta$. The behavior is consistent with classical Bayesian learning theory in the i.i.d. case, now extended to structured and dependent sequences. This correspondence indicates that predictive information faithfully reflects the effective degrees of freedom of the generative process.

**Beyond Finite Parametric Models.** When the data-generating process depends on an infinite or un-bounded set of latent parameters [7, 4], predictive information may vanish or grow sub-logarithmically, indicating insufficient structure for reliable forecasting or a mismatch between model and data complexity. In such cases, structural assumptions or inductive biases may be necessary to enable generalization and avoid overfitting.

### 4.3 Link Between Learning Curve and Minimal Achievable Risk

We now show how the universal learning curve $\Lambda(k)$ provides an upper bound on how close a model of order $k$ can get to the optimal regression risk.

**Proposition 4.5.** *For any $k \in \mathbb{N}$ and any $Q \in \mathcal{H}_k$,*

$$\mathcal{R}^\infty(Q^*) \; \leq \; \mathcal{R}^k(Q) - \Lambda(k).$$

*where $\mathcal{R}^\infty(Q^*)$ is the minimal risk achievable by the optimal predictor $Q^* = P(X_{t+1} \mid \mathbf{X}_{past})$.*
*Proof in Appendix A.11.*

In practice, the optimal achievable risk is given by $\mathcal{R}^\infty(Q^*) = \lim_{k \to \infty} H(X_{t+1} \mid \mathbf{X}_{t-k+1}^t)$. This notion parallels the Bayesian risk in classification tasks [50], where the focus lies on the gap between the current and optimal losses, particularly in online learning contexts [26, 27, 35].

Here, we aim to estimate $\mathcal{R}^\infty(Q^*)$ directly. The previous proposition shows that no model with finite memory $k$ can attain the optimal risk unless $\Lambda(k)$ is negligible. The term $\Lambda(k)$ quantifies the irreducible uncertainty due to limited context and serves as a data-dependent upper bound on the possible reduction in risk achievable by optimizing over $\mathcal{H}_k$.

Since the true risk $\mathcal{R}^k(Q)$ is not directly observable, we use its empirical estimate,

$$\hat{\mathcal{R}}_n^k(Q) = \frac{1}{n} \sum_{i=1}^n - \ln Q \left( X_{i,t+1} \mid \mathbf{X}_{i,t-k+1}^t \right). \tag{7}$$

To control the deviation between the true and estimated risks, we leverage Rademacher complexity for stationary sequences [36], together with standard concentration inequalities.

**Proposition 4.6.** *Assume $(X_t)_{t \in \mathbb{Z}}$ is a stationary process satisfying the conditions of [36]. Then, for any $\delta \in (0,1)$, with probability at least $1 - \delta$ over an $n$-sample drawn from the process, every $Q \in \mathcal{H}$ satisfies*

$$\mathcal{R}^\infty(Q^*) \leq \hat{\mathcal{R}}^k(Q) - \Lambda(k) + 2\widehat{\Re}_n(\mathcal{F}_k) + 3\frac{\ln(1/\delta)}{n}$$

*where*

$$\widehat{\Re}_n(\mathcal{F}_k) \; = \; \mathbb{E}_\sigma \left[ \sup_{f \in \mathcal{F}_k} \frac{1}{n} \sum_{i=1}^n \sigma_i \, f\left( \mathbf{X}_{i,0}^t, X_{i,t+1} \right) \right]$$

*is the empirical Rademacher complexity of the class $\mathcal{F}_k$ computed on the sample, with $\sigma_1, \ldots, \sigma_n$ i.i.d. Rademacher variables.*
*Proof in Appendix A.11.*

This proposition provides a formal bound on the best achievable risk. For simplicity, we assume $\hat{\mathcal{R}}_n^k(Q) = \hat{\mathcal{R}}^k(Q)$ throughout the rest of the paper to reduce notational complexity. Pre-trained models on various benchmark datasets are widely available in the literature. We argue that such models can be leveraged to estimate both the optimal regression order and the best achievable risk on the dataset they were trained on.

**Corollary 4.7.** *Suppose we have access to a trained model $Q_k \in \mathcal{H}_k$ for each regression order $k = 1, \ldots, M$. Then:*

$$\text{(i)} \quad \hat{\mathcal{R}}^\infty(Q^*) = \min_{1 \leq k \leq M} \{\hat{\mathcal{R}}^k(Q_k) - \Lambda(k)\},$$

$$\text{(ii)} \quad k^* \leq \arg \min_{1 \leq k \leq M} \{\hat{\mathcal{R}}^k(Q_k) - \Lambda(k)\}.$$

(i) *provides a way to estimate the minimal achievable risk.* (ii) *allows one to infer the optimal regression order from this same model collection.*
*Proof in Appendix A.12.*

If model performance plateaus beyond a certain context length $k$—that is, if the loss ceases to improve while $\Lambda(k)$ remains strictly positive—then the oracle reveals a lower achievable loss than what the model currently attains. This discrepancy indicates the presence of residual predictive structure that the model fails to capture, thereby offering a principled target for improvement. Oracle-based analysis thus provides a framework for assessing whether a model fully exploits the predictive information present in the data. Unlike `EvoRate` [55] or the forecastability measure [23], our estimator is model-dependent, as it relies on the hypothesis class $\mathcal{H}_k$. While this introduces sensitivity to the model architecture, it also enables direct comparisons between the estimated oracle risk and the empirical performance of candidate models, offering a more actionable diagnostic.

Finally, under a Mean Squared Error (MSE) loss and assuming the predictive distribution is multivariate Gaussian, it holds that $\mathcal{L}_{\text{mle}} = \mathcal{L}_{\text{mse}} + \text{const}$ [55]. This equivalence ensures that our framework remains applicable in standard Gaussian regression settings, thereby enhancing its practical utility.

## 5   Experiment

Throughout this section, we use empirical estimators of the predictive information $\mathbf{I}_{\text{pred}}$, following the procedure outlined in Appendix B. For clarity, we will refer to our estimate simply as $\mathbf{I}_{\text{pred}}$ in the remainder of this section. In the first part, we explain the estimation procedure for $\mathbf{I}_{\text{pred}}$, and then use it to derive the estimated learning curve $\hat{\Lambda}$ and the optimal risk $\hat{\mathcal{R}}^\infty(Q^*)$.

### 5.1   Estimation of $\mathbf{I}_{\text{pred}}$ on a Gaussian process

We consider the process $\{X_t\}_{t-k+1}^{t+k'}$, a $d$-dimensional Gaussian process with independent and identically distributed (i.i.d.) components. In this setting, it can be shown that

$$\mathbf{I}_{\text{pred}}(k, k') = \mathbf{I}(\mathbf{X}_{t-k+1}^t; \mathbf{X}_{t+1}^{t+k'}) = \frac{d}{2} \ln \left( \frac{|\Sigma_1^{(1)}||\Sigma_2^{(1)}|}{|\Sigma^{(1)}|} \right) \tag{8}$$

where $\Sigma_1$ denotes the covariance matrix of $\mathbf{X}_{t-k+1}^t$, $\Sigma_2$ the covariance of $\mathbf{X}_{t+1}^{t+k'}$, and $\Sigma$ the joint covariance matrix of the full sequence $\{X_t\}_{t-k+1}^{t+k'}$. [2]

Figure 1 illustrates the estimation results of $\mathbf{I}_{\text{pred}}(k, k')$ across various input dimensions and kernel types detailed in Appendix B.2. For this experiment, we fixed $k = 5$ and $k' = 10$. $\mathbf{I}_{\text{pred}}$−`True` denotes the theoretical value computed from Equation 8. Additional combinations of $(k, k')$ and their corresponding estimation results are provided in Appendix B.3.

---

[2]The superscript $(1)$ indicates that, due to the i.i.d. nature of the dimensions, the computation can be performed on the first dimension only and scaled by $d$.

| | AR (d=1) | AR (d=5) | AR (d=10) | Gaussian ($\rho=0.5$, d=1) | Gaussian ($\rho=0.5$, d=10) | Gaussian ($\rho=0.5$, d=20) | Gaussian ($\rho=0.8$, d=1) | Gaussian ($\rho=0.8$, d=10) | Gaussian ($\rho=0.8$, d=20) | Local-P (d=1) | Matérn 3/2 (d=1) | Matérn 3/2 (d=5) | Matérn 3/2 (d=10) | Matérn 5/2 (d=1) | Matérn 5/2 (d=5) | Matérn 5/2 (d=10) | Periodic (d=1) | RBF (d=1) | RQ (d=5) | RQ (d=10) | RQ (Extra) |
|---|---|---|---|---|---|---|---|---|---|---|---|---|---|---|---|---|---|---|---|---|---|
| $\hat{\mathbf{I}}_{\text{pred}}$-DV | 0.5 | 2.3 | 3.6 | 0.14 | 1.3 | 1.5 | 0.46 | 3.9 | 3.9 | 5 | 0.59 | 2.5 | 4.1 | 0.97 | 4 | 5.7 | 4.5 | 3.2 | 1.5 | 5.5 | 4.8 |
| $\hat{\mathbf{I}}_{\text{pred}}$-NWJ | 0.5 | 2.2 | 3.5 | 0.14 | 1.2 | 1.8 | 0.49 | 3.4 | 3.9 | 4.1 | 0.59 | 2.6 | 3.7 | 0.93 | 3.5 | 4.1 | 3.9 | 3.5 | 1.5 | 4.2 | 4.1 |
| $\hat{\mathbf{I}}_{\text{pred}}$-SMILE | 0.49 | 2.3 | 2.9 | 0.14 | 1.3 | 1.2 | 0.49 | 2.9 | 2.8 | 3.8 | 0.59 | 2.6 | 3.1 | 0.98 | 2.7 | 3.1 | 3.8 | 2.4 | 1.4 | 2.8 | 2.9 |
| $\hat{\mathbf{I}}_{\text{pred}}$-InfoNCE | 0.48 | 2.2 | 3.6 | 0.14 | 1.2 | 1.3 | 0.44 | 4.3 | 4.4 | 4.5 | 0.58 | 2.6 | 4.4 | 0.93 | 3.9 | 5.9 | 1.7 | 2.7 | 1.4 | 7.9 | 5.4 |
| $\hat{\mathbf{I}}_{\text{pred}}$-TUBA | 0.5 | 2.2 | 3.1 | 0.14 | 1.2 | 1.5 | 0.5 | 3 | 2.7 | 3.5 | 0.59 | 2.6 | 3.5 | 0.96 | 2.9 | 2.9 | 3.4 | 3.1 | 1.4 | 2.9 | 2 |
| $\mathbf{I}_{\text{pred}}$-True | 0.51 | 2.5 | 5.1 | 0.14 | 1.4 | 2.9 | 0.51 | 5.1 | 10 | 8.6 | 0.62 | 3.1 | 6.2 | 1 | 5 | 10 | 12 | 6.3 | 1.6 | 16 | 8 |

Figure 1: Estimation of $\mathbf{I}_{\text{pred}}(k, k')$ using various neural-based methods. Color encodes the estimation bias: blue regions indicate negative bias (underestimation), while the intensity reflects the magnitude of this bias.

While all methods provide relatively accurate estimates in low-dimensional settings (dimensionality $d \leq 20$), their performance deteriorates as the input space becomes more complex. Notably, methods like $\hat{\mathbf{I}}_{\text{pred}}$-SMILE and $\hat{\mathbf{I}}_{\text{pred}}$-NWJ exhibited consistent underestimation, in structured, high-dimensional regimes (e.g., Periodic and RQ kernels). Although these estimators inherently introduce variance and may lead to some instability in the results, refining the estimation of $\mathbf{I}_{\text{pred}}$ lies beyond the scope of this work and would warrant a dedicated investigation.

## 5.2 Autoregressive Process

We demonstrate the relevance of using predictive information $\mathbf{I}_{\text{pred}}$ to approximate the universal learning curve $\Lambda(k)$, as stated in Proposition 4.1. This experiment also serves to validate the effectiveness of $\Lambda(k)$ in identifying the true memory length of a Markovian process, in accordance with Proposition 4.2. To this end, we simulate a stationary vector autoregressive process $\{X_t\}_{t=0}^{N-1} \subset \mathbb{R}^3$ of order $p \in \{5, 10\}$. The initial states $X_0, \ldots, X_{p-1}$ are drawn independently from a standard multivariate normal distribution $\mathcal{N}(0, I_3)$. For $t \geq p$, the process evolves according to:

$$X_t = \frac{\rho}{p} \sum_{j=t-p}^{t-1} X_j + \sqrt{1-\rho^2}\, \epsilon_t, \tag{9}$$

where $\epsilon_t \sim \mathcal{N}(0, I_3)$ and the parameter $\rho \in (0, 1)$ controls the strength of temporal dependence.

Figure 2 compares the estimated learning curve $\hat{\Lambda}(k)$—obtained via the estimator of predictive information from Proposition 4.1—with a reference curve $\tilde{\Lambda}(k)$ computed from the known data-generating distribution. Although this theoretical curve is not available in real-world scenarios, it serves here as a useful benchmark. Since the full distribution is known, we can accurately approximate the entropy and thereby the true learning curve (see Appendix C for derivation details).

The results confirm that $\hat{\Lambda}(k)$ closely tracks the theoretical curve and successfully identifies the correct model order $k = p$. This supports both the statistical consistency of the estimator and the practical usefulness of the learning curve for model selection. Despite minor fluctuations due to estimation variance, the method exhibits a sharp transition at the correct order $k = p$, underscoring its robustness and precision in capturing the underlying temporal structure.

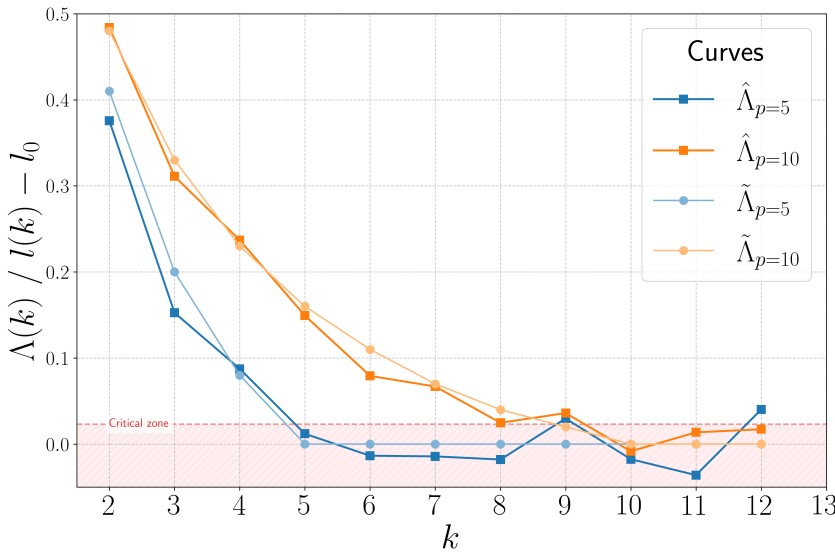

Figure 2: Learning curves $\Lambda(k)$ for AR processes with orders $p = 5$ and $p = 10$.

**Remark 5.1.** *The "Critical zone" designates the range where the universal learning curve $\Lambda(k)$ falls below a small threshold (here set at $0.02$). Because estimation does not yield exact zeros due to inherent variance, it is difficult to pinpoint the exact context length at which the curve vanishes. Introducing such a cutoff therefore provides a practical criterion: values of $\Lambda(k)$ below this level are considered effectively zero. While the threshold is somewhat arbitrary, it is chosen small relative to the typical magnitude of $\Lambda(k)$, yet large enough to remain robust to estimation noise.*

### 5.3 Estimating $\mathcal{R}^\infty(Q^*)$ for Ising Spin Sequences

Let $\mathbf{X}_t^T = \{X_u\}_{u=t}^T \in \mathcal{X}^{T-t+1}$ denote a sequence of binary variables where $\mathcal{X} = \{-1, +1\}$. We generate a sequence of length $T = 10{,}000{,}000$ starting with $X_0 \sim \text{Uniform}\{-1, +1\}$. The sequence is divided into blocks of size $M$. For each block, we sample $J \sim \mathcal{N}(0, 1)$ and evolve the sequence according to:

$$P(X_i = +1 \mid X_{i-1}, J) = \frac{\exp(JX_{i-1})}{\exp(JX_{i-1}) + \exp(-JX_{i-1})}. \tag{10}$$

This yields a piecewise-stationary Markov chain resembling a blockwise-random Ising process. We train both an MLP and an LSTM to predict $X_{t+1}$ from the $k$ past observations. Since an efficient predictor only requires a single parameter (logistic regression suffices), we fix $\dim \Theta = 1$, and use cross-entropy as the loss. For each block size $M \in \{10{,}000,\ 100{,}000,\ 1{,}000{,}000,\ 10{,}000{,}000\}$, we evaluate, for $1 \le k \le n$ (with $n = 19$), `EvoRate(k)`, along with our model-dependent estimates of the minimal achievable risk and the minimal loss empirically attained by LSTM and MLP models.

| $M$ | `EvoRate(10)` | $\hat{\mathcal{R}}_{\text{lstm}}^\infty(Q^*)$ | $\hat{\mathcal{R}}_{\text{mlp}}^\infty(Q^*)$ | $\min_{1 \le k \le n} \hat{\mathcal{R}}_{\text{lstm}}^k(Q)$ | $\min_{1 \le k \le n} \hat{\mathcal{R}}_{\text{mlp}}^k(Q)$ |
|---|---|---|---|---|---|
| 10,000 | 0.2758 | 0.3724 | 0.3719 | 0.4853 | 0.4872 |
| 100,000 | 0.2861 | 0.3684 | 0.3720 | 0.4679 | 0.4668 |
| 1,000,000 | 0.3269 | 0.3569 | 0.3390 | 0.3798 | 0.3660 |
| 10,000,000 | 0.4760 | 0.0697 | 0.0867 | 0.0733 | 0.0903 |

Table 1: Estimated minimal achievable risk and estimated minimal reached risk versus `EvoRate`.

As $M$ increases, the data becomes less complex: for $M = 10{,}000{,}000$, the coupling $J$ remains fixed, and the process reduces to a first-order Markov chain. Consequently, `EvoRate` increases, reflecting stronger underlying structure, and models achieve lower prediction losses. However, while `EvoRate` signals learnability, it does not provide a quantitative target, unlike our estimator $\hat{\mathcal{R}}^\infty(Q^*)$. Despite being model-dependent, the oracle estimates $\hat{\mathcal{R}}^\infty(Q^*)$ remain consistent across

LSTM and MLP, with only minor differences relative to their scale. It also aligns qualitatively with EvoRate: lower estimated risk corresponds to more predictable sequences. Importantly, it enables direct comparison with actual model performance. For example, when $M = 10,000$, the ratio $\hat{\mathcal{R}}^k(Q)/\hat{\mathcal{R}}^\infty(Q^*) \approx 1.3$ indicates suboptimal predictions, likely due to high non-stationarity. As $M$ grows, this ratio approaches 1, confirming improved model adequacy. Lastly, negative $\Lambda(k)$ estimates for $M = 10,000,000$ stem from instability (*see Table 8*) in $\hat{\Lambda}(k)$ when $k \gg p$ for a true order-$p$ Markov process (*see Figure 2*). Refining this estimator is necessary to avoid misinterpretation in low-complexity settings.

**Additional Remarks on Result Interpretability**

We estimate the parameter dimension $\dim(\Theta)$ using Corollary 4.4, with the true value being **1**. For $k = 10$, the estimator yields $\hat{p} = 2 \times k \times \hat{\Lambda}(k) = 0.9580$ when $M = 10,000$. Comparable values are obtained for other sample sizes $M$, indicating the consistency of the procedure in the parametric regime. In contrast, when $M = 10,000,000$, which corresponds to the chain length, the process effectively becomes Markovian and thus departs from the parametric setting. As theoretically expected, the estimator then approaches zero; for example, at $k = 10$ we obtain $\hat{p} = 0.0720$.

We can also evaluate the optimal regression orders $k^*$ for each model, which remain consistent across both, evolving together and staying within the same order of magnitude as $M$ increases.

| $M$ | $k^*_{\text{LSTM}}$ | $k^*_{\text{MLP}}$ |
|---|---|---|
| 10,000 | 1 | 1 |
| 100,000 | 1 | 1 |
| 1,000,000 | 18 | 16 |
| 10,000,000 | 10 | 9 |

Table 2: Estimated optimal regression orders.

# 6   Conclusion

This work addresses two fundamental questions in sequential modeling: **(i)** what is the minimal achievable risk when modeling sequential data, and **(ii)** is poor predictive performance due to model limitations or to the intrinsic unpredictability of the data?

To answer these questions, we introduced a unified information-theoretic framework centered around the learning curve $\Lambda(k)$, which quantifies the gain in predictive accuracy when extending the context from $k$ to $k + 1$. This quantity provides a principled way to connect statistical dependencies in the data to achievable predictive performance. Building on this foundation, we proposed a practical estimator of the minimal achievable risk, $\hat{\mathcal{R}}^\infty(Q^*)$ which bounds from below the performance of any predictor operating on the same data. This estimator enables a direct diagnostic test for model adequacy: if the empirical risk $\hat{\mathcal{R}}^k(Q)$ approaches $\hat{\mathcal{R}}^\infty(Q^*)$, then the observed performance is close to the intrinsic unpredictability of the process—suggesting that increasing model capacity or context length is unlikely to yield further improvements. Conversely, a significant gap between these two quantities reveals that the model underfits the data, pointing to unexploited temporal dependencies.

Through theoretical analysis, we demonstrated that $\Lambda(k)$ admits explicit asymptotic forms in both parametric and Markov regimes, linking the decay of the learning curve to intrinsic properties such as the parameter dimensionality or the Markov order. Empirical validation on controlled synthetic datasets confirmed these predictions: our estimator consistently recovered the true minimal risk, accurately distinguished between parametric and non-parametric regimes, and correctly identified whether performance limitations arose from model capacity or from the intrinsic randomness of the process.

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

# Appendix

# A  Proofs

## A.1  On the main properties of $\mathbf{I}_{\mathrm{pred}}$

**Proposition A.1** (Elementary properties of the predictive mutual information)**.** *For integers $k \geq 1$ and $k' \geq 1$, let*

$$\mathbf{I}_{\mathrm{pred}}(k, k') \;=\; \mathbf{I}\big(\mathbf{X}_{t-k+1}^{t} \,;\, \mathbf{X}_{t+1}^{t+k'}\big)$$

*denote the mutual information between a past block of length $k$ and a future block of length $k'$. Assuming that the process $(X_t)_{t \in \mathbb{Z}}$ is stationary (and therefore time–translation invariant), $\mathbf{I}_{\mathrm{pred}}$ satisfies:*

1.  ***Non-negativity.*** $\mathbf{I}_{\mathrm{pred}}(k, k') \geq 0$.

2.  ***Symmetry.*** $\mathbf{I}_{\mathrm{pred}}(k, k') = \mathbf{I}\big(\mathbf{X}_{t+1}^{t+k'} \,;\, \mathbf{X}_{t-k+1}^{t}\big)$.

3.  ***Monotonicity.***

    (a) *(Increasing past) For fixed $k'$, $\mathbf{I}_{\mathrm{pred}}(k + 1, k') \;\geq\; \mathbf{I}_{\mathrm{pred}}(k, k')$.*
    (b) *(Increasing future) For fixed $k$, $\mathbf{I}_{\mathrm{pred}}(k, k' + 1) \;\geq\; \mathbf{I}_{\mathrm{pred}}(k, k')$.*

4.  ***Chain-rule decomposition.*** *For any split $k_1 + k_2 = k'$,*

    $$\mathbf{I}_{\mathrm{pred}}(k, k_1 + k_2) \;=\; \mathbf{I}_{\mathrm{pred}}(k, k_1) + \mathbf{I}\big(\mathbf{X}_{t-k+1}^{t} \,;\, \mathbf{X}_{t+k_1+1}^{t+k_1+k_2} \,\big|\, \mathbf{X}_{t+1}^{t+k_1}\big),$$

    *and an analogous identity holds when the past block is partitioned.*

5.  ***Data-processing inequality.*** *For any measurable maps $f$ and $g$,*

    $$\mathbf{I}\big(f(\mathbf{X}_{t-k+1}^{t}) \,;\, g(\mathbf{X}_{t+1}^{t+k'})\big) \;\leq\; \mathbf{I}_{\mathrm{pred}}(k, k').$$

6.  ***Convergence to excess entropy.*** *Let $E$ denote the excess (or predictive) entropy of the process: $E := \mathbf{I}\big(\mathbf{X}_{-\infty}^{t} \,;\, \mathbf{X}_{t+1}^{\infty}\big)$. Then $\lim\limits_{k, k' \to \infty} \mathbf{I}_{\mathrm{pred}}(k, k') = E$.*

*Proof.* All items follow directly from classical properties of Shannon's mutual information (see, e.g., [13, Chs. 2–3]).

- *(1)–(2)* are immediate from non-negativity and symmetry of $I(X; Y)$.

- *(3)* Extending the conditioning set can only reduce conditional entropy, hence cannot decrease $I(X; Y)$; apply this with $X = \mathbf{X}_{t-k}^{t}$ (resp. $\mathbf{X}_{t-k+1}^{t}$) and $Y = \mathbf{X}_{t+1}^{t+k'}$ (resp. $\mathbf{X}_{t+1}^{t+k'+1}$).

- *(4)* is the chain rule $I(X; YZ) = I(X; Y) + I(X; Z \mid Y)$ applied to suitably chosen blocks.

- *(5)* is the data-processing inequality: applying measurable maps cannot increase mutual information.

- *(6)* For fixed $k$, $\mathbf{I}_{\mathrm{pred}}(k, k')$ is non-decreasing and bounded above by $E$; likewise in $k$. Monotone convergence plus stationarity gives the limit $E$.

$\square$

**Proposition A.2** (Convergence of $\mathbf{I}_{\mathrm{pred}}$ when the future window grows)**.** *Let $(X_t)_{t \in \mathbb{Z}}$ be a stationary process with finite entropy rate $h_0 := \lim_{n \to \infty} H(X_1^n)/n$. Write the block entropy as*

$$H(n) = n\, h_0 + H_1(n), \qquad n \geq 1,$$

*where the* sub-extensive term *satisfies $H_1(n) = o(n)$. Then, for every fixed $k \geq 1$,*

$$\lim_{k' \to \infty} \mathbf{I}_{\mathrm{pred}}(k, k') \;=\; H_1(k).$$

*Proof.* For any $k, k' \geq 1$ the predictive mutual information can be written as

$$\mathbf{I}_{\text{pred}}(k, k') \ = \ H(k) + H(k') - H(k + k').$$

Insert the decomposition $H(n) = n h_0 + H_1(n)$:

$$\mathbf{I}_{\text{pred}}(k, k') = \big[ k h_0 + H_1(k) \big] + \big[ k' h_0 + H_1(k') \big] - \big[ (k + k') h_0 + H_1(k + k') \big] = H_1(k) + H_1(k') - H_1(k + k').$$

Now let $k' \to \infty$ while keeping $k$ fixed. Because $H_1(n) = o(n)$, the difference $H_1(k') - H_1(k + k')$ vanishes:

$$|H_1(k') - H_1(k + k')| \ \leq \ o(k') + o(k + k') = o(k'),$$

hence tends to $0$. Therefore $\lim_{k' \to \infty} \mathbf{I}_{\text{pred}}(k, k') = H_1(k)$. $\qquad\square$

**Proposition A.3** (Asymptotic equivalence between $H(k)$ and $\mathbf{I}_{\text{pred}}$). *Assume Hypothesis* $(\mathbf{H_0})$ *(stationarity with finite entropy rate $l_0$ and sub-extensive remainder $H_1$). Fix a non-decreasing sequence $k' = k'(k)$ such that $k' \geq k$ and $k' = \mathcal{O}(k)$ when $k \to \infty$. Then*

$$H\big(\mathbf{X}_{t-k+1}^{t}\big) \ = \ k\, l_0 + \mathbf{I}_{\text{pred}}(k, k') + o(k), \qquad k \to \infty,$$

*hence in particular* $\mathbf{I}_{\text{pred}}(k, k')/k \longrightarrow 0$.

*Proof.* Write the block entropy as $H(n) = n\, l_0 + H_1(n)$ with $H_1(n) = o(n)$. Using the identity $\mathbf{I}_{\text{pred}}(k, k') = H(k) + H(k') - H(k + k')$ and the decomposition above gives

$$\mathbf{I}_{\text{pred}}(k, k') = H_1(k) + H_1(k') - H_1(k + k').$$

Because $k' \geq k$ and $k' = \mathcal{O}(k)$, sub-extensiveness yields $H_1(k') - H_1(k + k') = o(k)$. Therefore

$$\mathbf{I}_{\text{pred}}(k, k') = H_1(k) + o(k).$$

Insert this into $H(k) = k\, l_0 + H_1(k)$ to obtain $H(k) = k\, l_0 + \mathbf{I}_{\text{pred}}(k, k') + o(k)$. Finally, $\mathbf{I}_{\text{pred}}(k, k')/k = H_1(k)/k + o(1) \xrightarrow{k \to \infty} 0$ because $H_1(k) = o(k)$. $\qquad\square$

**Proposition A.4** (Exact link between EvoRate and $\mathbf{I}_{\text{pred}}$). *Assume the process is stationary and time–translation invariant. For any integers $k \geq 1$ and $k' \geq 1$,*

$$\mathbf{I}_{\text{pred}}(k, k') \ = \ \texttt{EvoRate}(k) \ + \ \big[ H_1(k') - H_1(1) \big] \ + \ \big[ H_1(k + 1) - H_1(k + k') \big],$$

*where the block entropy is decomposed as $H(n) = n\, l_0 + H_1(n)$ with a sub-extensive part $H_1(n) = o(n)$. In particular, for a single-step future window ($k' = 1$) we recover*

$$\mathbf{I}_{\text{pred}}(k, 1) \ = \ \texttt{EvoRate}(k).$$

*Proof.* By stationarity, $\mathbf{I}_{\text{pred}}(k, k') = H_1(k') + H_1(k) - H_1(k + k')$. The evolutionary rate is defined as $\texttt{EvoRate}(k) = H_1(1) + H_1(k) - H_1(k + 1)$. Solving the last equation for $H_1(k)$ gives $H_1(k) = \texttt{EvoRate}(k) - H_1(1) + H_1(k + 1)$. Substituting this into the expression of $\mathbf{I}_{\text{pred}}$ yields

$$\mathbf{I}_{\text{pred}}(k, k') = \texttt{EvoRate}(k) + \big[ H_1(k') - H_1(1) \big] + \big[ H_1(k + 1) - H_1(k + k') \big],$$

which is the desired identity. Setting $k' = 1$ cancels both bracketed terms, proving the special case $\mathbf{I}_{\text{pred}}(k, 1) = \texttt{EvoRate}(k)$. $\qquad\square$

## A.2 Learning theory

**Proposition A.5** (Predictive information versus the cumulative learning curve). *Assume ($H_0$) (strict stationarity and a finite entropy rate). For $n \geq 1$ write the block entropy as $H(n) = n\, \ell_0 + H_1(n)$, where the* sub-extensive *term satisfies $H_1(n) = o(n)$. Define the order–$k$ entropy rate $\ell(k) = H(k + 1) - H(k)$ and the* universal learning curve $\Lambda(k) = \ell(k) - \ell_0 = H_1(k + 1) - H_1(k)$.

*Then, for all integers $k \geq 1$ and $k' \geq k$,*

$$\sum_{i=1}^{k} \Lambda(i) \ - \ H_1(k) \ \leq \ \mathbf{I}_{\text{pred}}(k, k') \ \leq \ \sum_{i=1}^{k} \Lambda(i). \tag{11}$$

*Moreover*

$$k\, \Lambda(k) \ \xrightarrow[k \to \infty]{} \ 0, \tag{12}$$

*so the width of the sandwich in (11) is $H_1(k) = o(k)$.*

*Proof.* From the decomposition $H(n) = n\ell_0 + H_1(n)$ we have $\Lambda(k) = H_1(k+1) - H_1(k)$ and $S_k := \sum_{i=1}^{k} \Lambda(i) = H_1(k+1) - H_1(1)$.

For any $k' \geq k$,

$$\mathbf{I}_{\text{pred}}(k, k') = H(k) + H(k') - H(k+k') = H_1(k) + H_1(k') - H_1(k+k').$$

Hence

$$\mathbf{I}_{\text{pred}}(k, k') = S_k - \left[ H_1(k+k') - H_1(k') \right]. \tag{13}$$

Because $H_1$ is non-decreasing, $H_1(k+k') \geq H_1(k')$; the bracket in (13) is therefore non-negative and we obtain the upper bound $\mathbf{I}_{\text{pred}}(k, k') \leq S_k$.

Sub-additivity of $H_1$ gives $H_1(k+k') \leq H_1(k') + H_1(k)$, hence $H_1(k+k') - H_1(k') \leq H_1(k)$. Inserting this into (13) yields the lower bound $\mathbf{I}_{\text{pred}}(k, k') \geq S_k - H_1(k)$. Together these inequalities establish (11).

Finally, $k\,\Lambda(k) = k\left[ H_1(k+1) - H_1(k) \right] \leq H_1(k+1) = o(k)$, which proves (12). $\qquad \square$

**Proposition A.6** (Predictive–information increment vs. universal learning curve). *Fix $k \geq 1$. Then, under hypothesis (**$H_0$**),*

$$\mathbf{I}_{\text{pred}}(k+1, k') \; - \; \mathbf{I}_{\text{pred}}(k, k') \; \xrightarrow[k' \to \infty]{} \; \Lambda(k),$$

*where $\Lambda(k) = \ell(k) - \ell_0$ is the universal learning curve and $\ell(k) = H(k+1) - H(k)$ denotes the order-$k$ entropy rate.*

*Proof.* Write the block entropy in its extensive–plus–remainder form $H(n) = n\,\ell_0 + H_1(n)$ with $H_1(n) = o(n)$. Consequently

$$\ell(k) = H(k+1) - H(k) = H_1(k+1) - H_1(k), \qquad \Lambda(k) = \ell(k) - \ell_0 = H_1(k+1) - H_1(k). \tag{1}$$

Proposition A.2 (convergence to the sub-extensive part) gives, for every fixed $m$,

$$\lim_{k' \to \infty} \mathbf{I}_{\text{pred}}(m, k') = H_1(m). \tag{2}$$

Applying (2) with $m = k$ and $m = k+1$ we obtain

$$\lim_{k' \to \infty} \left[ \mathbf{I}_{\text{pred}}(k+1, k') - \mathbf{I}_{\text{pred}}(k, k') \right] = H_1(k+1) - H_1(k).$$

Combined with (1), this equals $\Lambda(k)$, establishing the claimed convergence. $\qquad \square$

## A.3  Asymptotic behavior of $\mathbf{I}_{\text{pred}}$

### A.3.1  Markovian case

We believe that the main limitations of `EvoRate` stem from the fact that its limiting values are unclear. It is difficult to conclude about its limits. In contrast, the study of $\mathbf{I}_{\text{pred}}$ and its limiting values can provide meaningful insights and may help to uncover underlying patterns in sequential data. This is particularly true in the case of Markovian processes:

**Proposition A.7.** *Assume that $X$ is a Markov process of order $m$ and that $k' \geq k \geq m$ so that the relevant past information is contained in $\mathbf{X}_{t-m+1}^{t}$, then*

$$\mathbf{I}_{pred}(k, k') = I\left( \mathbf{X}_{t-k+1}^{t}, X_{t+1}^{t+k'} \right) = \mathbb{E}_{\mathbf{X}_{t-m+1}^{t+m}} \left[ \ln \frac{P\left( X_{t+1}^{t+m} \mid \mathbf{X}_{t-m+1}^{t} \right)}{P\left( X_{t+1}^{t+m} \right)} \right].$$

*In particular for a order-1 Markov process, $\mathbf{I}_{pred}(k, k') = $ `EvoRate`$(1)$.*

*Proof.* We start from the definition

$$\mathbf{I}_{\text{pred}}(k, k') = I\left( \mathbf{X}_{t-k+1}^{t}, X_{t+1}^{t+k'} \right) = \mathbb{E}_{\mathbf{X}_{t-k+1}^{t+k'}} \left[ \ln \frac{P\left( X_{t+1}^{t+k'} \mid \mathbf{X}_{t-k+1}^{t} \right)}{P\left( X_{t+1}^{t+k'} \right)} \right]$$

Assume that $X$ is a Markov process of order $m$ and that $k \geq m$ so that the relevant past information is contained in $\mathbf{X}_{t-m+1}^t$. We start by decomposing the unconditionnal probability (the denominator)

$$P\left(X_{t+1}^{t+k'}\right) = \prod_{j=1}^{k'} P\left(X_{t+j} \mid X_{t+1}^{t+j-1}\right)$$

Since the process is Markov of order $m$, the simplification by the Markov property holds only when there are at least $m$ prior observations in the sequence $X_{t+1}^{t+j-1}$. For $j = 1, \ldots, m$, the conditional probability $P\left(X_{t+j} \mid X_{t+1}^{t+j-1}\right)$ remains as is because $X_{t+1}^{t+j-1}$ contains fewer than $m$ observations (with the understanding that, by convention, for $j = 1$ we have $P\left(X_{t+1} \mid \emptyset\right) = P\left(X_{t+1}\right)$ ). For $j \geq m+1$, the Markov property yields $P\left(X_{t+j} \mid X_{t+1}^{t+j-1}\right) = P\left(X_{t+j} \mid X_{t+j-m}^{t+j-1}\right)$.

Thus, the full decomposition is

$$P\left(X_{t+1}^{t+k'}\right) = \underbrace{\prod_{j=1}^{m} P\left(X_{t+j} \mid X_{t+1}^{t+j-1}\right)}_{\text{non-simplified terms}} \times \underbrace{\prod_{j=m+1}^{k'} P\left(X_{t+j} \mid X_{t+j-m}^{t+j-1}\right)}_{\text{Markov terms}}. \tag{14}$$

Now, consider the numerator $P\left(X_{t+1}^{t+k'} \mid \mathbf{X}_{t-k+1}^t\right)$. Since $k \geq m$, the available past $\mathbf{X}_{t-k+1}^t$ contains at least the last $m$ values, i.e., $X_{t-m+1}^t$. We directly have by similar arguments,

$$P\left(X_{t+1}^{t+k'} \mid \mathbf{X}_{t-k+1}^t\right) = \prod_{j=1}^{k'} P\left(X_{t+j} \mid X_{t-m+1}^{t+j-1}\right) \tag{15}$$

Using 14 and 15 the factors for $j \geq m+1$ in the numerator and the denominator match and cancel each other in the ratio. In other words, the difference between the conditional probability $P\left(X_{t+1}^{t+k'} \mid \mathbf{X}_{t-k+1}^t\right)$ and the unconditional probability $P\left(X_{t+1}^{t+k'}\right)$ is confined to the first $m$ factors. We therefore write

$$\frac{P\left(X_{t+1}^{t+k'} \mid \mathbf{X}_{t-k+1}^t\right)}{P\left(X_{t+1}^{t+k'}\right)} = \frac{\prod_{j=1}^{m} P\left(X_{t+j} \mid X_{t-m+1}^{t+j-1}\right)}{\prod_{j=1}^{m} P\left(X_{t+j} \mid X_{t+1}^{t+j-1}\right)}$$

We obtain, after simplification :

$$\frac{P\left(X_{t+1}^{t+k'} \mid \mathbf{X}_{t-k+1}^t\right)}{P\left(X_{t+1}^{t+k'}\right)} = \frac{P\left(X_{t+1}^{t+m} \mid X_{t-m+1}^t\right)}{P\left(X_{t+1}^{t+m}\right)}$$

leading to

$$\mathbf{I}_{\text{pred}}\left(k, k'\right) = I\left(\mathbf{X}_{t-k+1}^t, X_{t+1}^{t+k'}\right) = \mathbb{E}_{\mathbf{X}_{t-m+1}^{t+m}}\left[\ln \frac{P\left(X_{t+1}^{t+m} \mid \mathbf{X}_{t-m+1}^t\right)}{P\left(X_{t+1}^{t+m}\right)}\right]$$

$\square$

**Proposition A.8.** *Let's suppose $\mathbf{X}_t^T$ is a Markovian process of order $m$. Then*

$$\forall k, \qquad k \geq m \Rightarrow \Lambda(k) = 0,$$

*Proof.* From the definition of the universal learning curve, and noting that $\mathbf{I}_{\text{pred}}$ is constant for $k$ greater than $m$. $\square$

### A.3.2 Predictive information for a parametrised stationary process

Our goal is to derive the asymptotic expansion of the predictive mutual information when the data–generating process belongs to a finite-dimensional parametric family. The argument extends Bialek and Tishby [7], who treated the i.i.d. case, to the setting of *dependent* sequences.

**Setup and notation.** Let $\{X_t\}_{t\in\mathbb{Z}}$ be a strictly stationary stochastic process taking values in a Polish space $\mathcal{X} \subset \mathbb{R}^d$. For integers $k \geq 1$ and $k' \geq k$ denote

$$\mathbf{X}_{\mathrm{past}} := \mathbf{X}_{t-k+1}^t = (X_{t-k+1},\dots,X_t), \qquad \mathbf{X}_{\mathrm{fut}} := \mathbf{X}_{t+1}^{t+k'} = (X_{t+1},\dots,X_{t+k'}).$$

Write $p(x_1^n)$ for the joint density of the block $\mathbf{X}_1^n := (X_1,\dots,X_n)$ with respect to a reference measure $\lambda^{d\otimes n}$ on $\mathcal{X}^n$.

**Parametric model.** Assume that there exists

- an *open* parameter set $\Theta \subset \mathbb{R}^p$ ($p < \infty$),
- a *prior density* $\mathcal{P}\colon \Theta \to (0,\infty)$ of class $C^1$ on $\Theta$,
- a *Kolmogorov-consistent* family of densities $\{Q_\theta^{(n)}\}_{\theta\in\Theta, n\geq 1}$ such that, for each $n \geq 1$,

$$p(x_1^n) = \int_\Theta Q_\theta^{(n)}(x_1^n)\, \mathcal{P}(\theta)\, d\theta.$$

Consistency means that $(Q_\theta^{(n)})_{n\geq 1}$ are the marginals of a single probability law $Q_\theta$ on $\mathcal{X}^\mathbb{Z}$ [28, Chap. 8]. The true parameter $\bar{\theta} \in \Theta$ is the (unknown) value generating the observations.

**Main regularity hypotheses.**

1. **Stationarity and geometric $\alpha$-mixing.** Under the true parameter $\bar{\theta}$, the process is strictly stationary and ergodic, with Rosenblatt mixing coefficients satisfying $\alpha_{\bar{\theta}}(n) \leq Ce^{-cn}$ for some constants $C, c > 0$.

2. **$C^3$ identifiability of the KL map.** The function $\theta \mapsto D_{\mathrm{KL}}(Q_\theta \,\|\, Q_{\bar{\theta}})$ is three-times continuously differentiable on a neighbourhood of $\bar{\theta}$, attains its unique minimum at $\bar{\theta}$, and has positive-definite Hessian $\mathcal{F}$ at that point.

3. **Finite entropy rate.** The block entropy $H(n) := H_{Q_{\bar{\theta}}}(\mathbf{X}_1^n)$ satisfies the Shannon–McMillan property $H(n) = n\,\ell_0 + o(n)$ with $\ell_0 < \infty$.

**Theorem A.9** (Asymptotics of the predictive mutual information). *Under assumptions 1–3, as $k \to \infty$ with $k' \geq k$,*

$$\mathbf{I}_{\mathrm{pred}}(k, k') = I(\mathbf{X}_{\mathrm{past}}; \mathbf{X}_{\mathrm{fut}})$$

$$= \frac{p}{2}\ln k + \frac{1}{2}\ln\det\mathcal{F} - \frac{p}{2}\ln(2\pi) + \ln\mathcal{P}(\bar{\theta}) + \mathcal{O}(k^{-1}). \tag{16}$$

*Consequently the universal learning curve $\Lambda(k) := \ell(k) - \ell_0$ obeys*

$$\Lambda(k) = \frac{p}{2k} - \frac{p}{4k^2} + \mathcal{O}(k^{-3}). \tag{17}$$

*All $o(\cdot)$ and $\mathcal{O}(\cdot)$ symbols are uniform in $\theta \in \mathcal{N}(\bar{\theta})$.*

*Proof.* Let $n := k + k'$. For $\theta \in \Theta$ set $L_n(\theta) := \ln(Q_\theta^{(n)}(\mathbf{X}_1^n)/Q_{\bar{\theta}}^{(n)}(\mathbf{X}_1^n))$. Exponential $\alpha$-mixing together with a Bernstein-type blocking argument [20, 19] yields a *uniform* strong law of large numbers:

$$\sup_{\theta\in\mathcal{N}(\bar{\theta})} \left| n^{-1}L_n(\theta) + D_{\mathrm{KL}}(Q_\theta\|Q_{\bar{\theta}}) \right| \xrightarrow[n\to\infty]{Q_{\bar{\theta}}\text{-a.s.}} 0. \tag{18}$$

Assumption 2 gives the quadratic expansion $D_{\mathrm{KL}}(Q_\theta\|Q_{\bar{\theta}}) = \frac{1}{2}(\theta - \bar{\theta})^\top \mathcal{F}(\theta - \bar{\theta}) + \mathcal{O}(\|\theta - \bar{\theta}\|^3)$. Plugging this into (18) and integrating, we split the marginal likelihood as

$$p(\mathbf{X}_1^n) = Q_{\bar{\theta}}^{(n)}(\mathbf{X}_1^n) \int_\Theta \exp(L_n(\theta))\, \mathcal{P}(\theta)\, d\theta =: Q_{\bar{\theta}}^{(n)}(\mathbf{X}_1^n)\, \mathcal{Z}_n.$$

*Laplace expansion.* Because $D_{\mathrm{KL}}(\theta\|\bar\theta) \geq c\|\theta - \bar\theta\|^2$ in $\mathcal{N}(\bar\theta)$, decompose $\Theta = B(\bar\theta, n^{-1/2+\delta}) \cup$ (complement). On the small ball, $L_n(\theta)$ is dominated by the quadratic form $-\frac{n}{2}(\theta - \bar\theta)^\top \mathcal{F}(\theta - \bar\theta)$ with a cubic remainder uniformly $o(1)$. Classical Laplace–Watson lemma [9, Chap. 6] yields

$$\mathcal{Z}_n = (2\pi)^{p/2}\, n^{-p/2}\, \frac{\mathcal{P}(\bar\theta)}{\sqrt{\det \mathcal{F}}}\, \big\{1 + \mathcal{O}(n^{-1})\big\}. \tag{19}$$

The contribution of the complement is $o\big(n^{-p/2}\big)$ because the integrand decays like $\exp\big[-cn^\delta\big]$.

*Block entropy decomposition.* Taking the $-\ln$ and the $Q_{\bar\theta}$-expectation of $Q_{\bar\theta}^{(n)}(\mathbf{X}_1^n)\,\mathcal{Z}_n$ and invoking 3 give

$$H(n) = n\,\ell_0 + \frac{p}{2}\ln n + \frac{1}{2}\ln\det\mathcal{F} - \frac{p}{2}\ln(2\pi) + \ln\mathcal{P}(\bar\theta) + \mathcal{O}(n^{-1}). \tag{20}$$

*Predictive information.* By definition $\mathbf{I}_{\mathrm{pred}}(k,k') = H(k) + H(k') - H(k+k')$. Substituting (20) with $n = k, k', k + k'$ cancels the extensive $n\ell_0$ parts; the remaining sub-extensive contributions yield (16). Finally

$$\Lambda(k) = \ell(k) - \ell_0 = H(k+1) - 2H(k) + H(k-1) = \frac{p}{2k} - \frac{p}{4k^2} + \mathcal{O}(k^{-3}),$$

where the last equality follows from a Taylor expansion $\ln(k+1) - \ln k = k^{-1} - \frac{1}{2}k^{-2} + O(k^{-3})$. Uniformity of the remainders is ensured by the uniform Laplace estimate (19). This completes the proof. $\qquad\square$

**Remark A.10.** *The leading term $\frac{p}{2}\ln k$ coincides with the* model-complexity penalty *in Bayesian minimum description length [12, 4]. The constant $\ln\mathcal{P}(\bar\theta) - \frac{p}{2}\ln(2\pi)$ depends on the local prior mass; it vanishes in the derivative $\Lambda(k)$ but is essential for the absolute scale of $\mathbf{I}_{\mathrm{pred}}$.*

## A.4 Minimal achievable risk

**Proposition A.11** (Learning–curve surplus bound)**.** *Let $(X_t)_{t\in\mathbb{Z}}$ be stationary and geometrically $\alpha$–mixing, i.e. $\alpha(n) \leq Ce^{-cn}$ for constants $C, c > 0$. For every regression order $k \in \mathbb{N}$, every predictor $Q \in \mathcal{H}_k$, every sample size $n \geq 1$ and every confidence $\delta \in (0,1)$, with probability at least $1 - \delta$*

$$\mathcal{R}^\infty(Q^*) \leq \hat{\mathcal{R}}^k(Q) - \Lambda(k) + 2\widehat{\Re}_n(\mathcal{F}_k) + 3\frac{\ln(1/\delta)}{n}, \tag{21}$$

*where $\widehat{\Re}_n(\mathcal{F}_k)$ is the empirical Rademacher complexity of the loss class $\mathcal{F}_k = \big\{ (\mathbf{x}, x') \mapsto -\ln Q(x' \mid \mathbf{x}) : Q \in \mathcal{H}_k \big\}$.*

*Proof.* For any $Q \in \mathcal{H}_k$

$$\mathcal{R}^k(Q) = \mathcal{R}^\infty(Q^*) + \big[\mathcal{R}^k(Q) - \mathcal{R}^k(Q^*)\big] + \Lambda(k),$$

because $\Lambda(k) = \mathcal{R}^k(Q^*) - \mathcal{R}^\infty(Q^*)$. Since the loss function $\ell(\cdot,\cdot)$ is non–negative, $\mathcal{R}^k(Q) \geq \mathcal{R}^k(Q^*)$, hence

$$\mathcal{R}^k(Q) \geq \mathcal{R}^\infty(Q^*) + \Lambda(k).$$

Rearranging gives the *excess-risk identity*

$$\mathcal{R}^\infty(Q^*) \leq \mathcal{R}^k(Q) - \Lambda(k). \tag{22}$$

Because the sequence is geometrically mixing, Theorem 2 of McDonald and Shalizi [36] (their bounded-loss Rademacher bound for $\alpha$–mixing processes) applies to $\mathcal{F}_k$: with probability $\geq 1 - \delta$

$$\mathcal{R}^k(Q) \leq \hat{\mathcal{R}}^k(Q) + 2\widehat{\Re}_n(\mathcal{F}_k) + 3\frac{\ln(1/\delta)}{n} \quad \text{for every } Q \in \mathcal{H}_k. \tag{23}$$

Insert (23) into the right–hand side of (22) to obtain (21).

$\square$

**Corollary A.12** (Oracle estimator and minimal order). *Assume the high-probability event of Proposition A.11. For a collection of pre-trained models $\{Q_k \in \mathcal{H}_k\}_{k=1}^M$ define*

$$\hat{\mathcal{R}}^\infty(Q^*) := \min_{1 \le k \le M} \{\hat{\mathcal{R}}^k(Q_k) - \Lambda(k)\}.$$

*Then*

$$\mathcal{R}^\infty(Q^*) \le \hat{\mathcal{R}}^\infty(Q^*) \quad and \quad k^\dagger \in \arg\min_{1 \le k \le M} \{\hat{\mathcal{R}}^k(Q_k) - \Lambda(k)\} \implies k^\dagger \ge k^*,$$

*where $k^* := \inf\{k : \Lambda(k) = 0\}$ is the minimal order for which the universal learning curve vanishes.*

*Proof.* Under the high-probability event, inequality (21) is valid for each $k = 1, \ldots, M$ when evaluated at $Q_k$. Discarding the non-negative complexity terms yields

$$\mathcal{R}^\infty(Q^*) \le \hat{\mathcal{R}}^k(Q_k) - \Lambda(k), \qquad \forall k \le M.$$

Taking the minimum over $k$ proves the claimed upper bound on $\mathcal{R}^\infty(Q^*)$. If $k^\dagger$ realises that minimum while $k^\dagger < k^*$, then $\Lambda(k^\dagger) > \Lambda(k^*) = 0$ and monotonicity of $\Lambda$ would contradict optimality of $k^\dagger$. Hence $k^\dagger \ge k^*$. $\qquad\square$

# B    Estimation of $\mathbf{I}_{\text{pred}}$

## B.1    Estimating $\mathbf{I}_{\text{pred}}$ with Variational Neural Estimators

We estimate the predictive mutual information $\mathbf{I}_{\text{pred}}$ using variational techniques based on recent advances in neural MI estimation. Given a $d$-dimensional time series $\{X_t\}$, our goal is to estimate

$$\mathbf{I}_{\text{pred}}(\mathbf{X}_{t-k+1}^t; \mathbf{X}_{t+1}^{t+k'}),$$

which quantifies the mutual information between a past window of $k$ time steps and a future window of $k'$ steps.

**Variational Estimators.**    Our approach builds on contrastive lower bounds of mutual information, including SMILE [52], NWJ [39], InfoNCE [40], TUBA [43], and DV [5]. These methods rely on a parameterized critic function $f_\theta(x, y)$, which scores pairs of past and future segments. The critic is trained to distinguish between positive pairs (sampled from the joint distribution $p(x, y)$) and negative pairs (from the product of marginals $p(x)p(y)$).

**Training Objective.**    The mutual information estimate is obtained by maximizing a variational objective of the form:

$$\mathbf{I}_{\text{pred}}(k, k') = \max_\theta \left\{ \frac{1}{B} \sum_{i=1}^B f_\theta(x_i, y_i) - \ln \left( \frac{1}{B(B-1)} \sum_{\substack{i,j=1 \\ i \ne j}}^B \exp(f_\theta(x_i, y_j)) \right) \right\}, \tag{24}$$

where $B$ is the batch size and the second term acts as a contrastive regularizer, penalizing high scores on mismatched pairs.

**Critic Architectures and Optimization.**    To capture the structure of temporal data, we experiment with multiple critic architectures: *Separable* (independent encodings for past and future), *Concatenated* (joint embeddings), and *Sequential* (LSTM-based encoders). The critic parameters $\theta$ are optimized using the Adam optimizer with stochastic gradient updates.

**Data Sampling.**    Training batches are constructed either by sampling synthetic data from known generators, or by extracting context–future pairs from long, continuous sequences. The full training procedure is detailed in Algorithm E.

## B.2 Synthetic data: kernel-based methods

For this part, we don't assume the invariance of temporal translation and stationarity of the process, as our concern is to verify that the estimators are working correctly.

**Proposition B.1.** *(Theoretical value of $\mathbf{I}_{pred}$) In the particular case where we consider $\{X_t\}_{t-k+1}^{t+k'}$ a Gaussian process of dimension $d$ with all dimensions independent and of the same distribution, we can compute explicitly $\mathbf{I}_{pred}$.*

$$\mathbf{I}_{pred}(k, k') = \mathbf{I}(\mathbf{X}_{t-k+1}^t, X_{t+1}^{t+k'}) = \frac{d}{2} \ln \left( \frac{\left|\Sigma_1^{(1)}\right| \left|\Sigma_2^{(1)}\right|}{\left|\Sigma^{(1)}\right|} \right).$$

*Where $\Sigma_1$ represents the covariance matrix of $\mathbf{X}_{t-k+1}^t$, $\Sigma_2$ the covariance matrix of $X_{t+1}^{t+k'}$ and $\Sigma$ the joint covariance matrix. The index $(1)$ means that we can just look at the first dimension of the Gaussian process.*

*Proof.* For a Gaussian process with dimension $d$ and independent dimensions, the predictive information formula is:

$$I(X_{\text{past}}; X_{\text{future}}) = \sum_{j=1}^d \frac{1}{2} \ln \left( \frac{|\Sigma_1^{(j)}||\Sigma_2^{(j)}|}{|\Sigma^{(j)}|} \right) \tag{25}$$

Using the differential entropy property of a Gaussian vector $X$ of dimension $n$ with covariance matrix $\Sigma$:

$$h(X) = \frac{1}{2} \ln((2\pi e)^n |\Sigma|) \tag{26}$$

Due to the independence of dimensions, the mutual information decomposes as:

$$I(X_{\text{past}}; X_{\text{future}}) = \sum_{j=1}^d I(X_{\text{past}}^{(j)}; X_{\text{future}}^{(j)}) \tag{27}$$

For each dimension $j$, with $\Sigma_1^{(j)}$ representing the covariance matrix of $X_{\text{past}}^{(j)}$, $\Sigma_2^{(j)}$ of $X_{\text{future}}^{(j)}$, and $\Sigma^{(j)}$ the joint covariance matrix:

$$I(X_{\text{past}}^{(j)}; X_{\text{future}}^{(j)}) = h(X_{\text{past}}^{(j)}) + h(X_{\text{future}}^{(j)}) - h(X_{\text{past}}^{(j)}, X_{\text{future}}^{(j)}) \tag{28}$$

$$= \frac{1}{2} \ln((2\pi e)^{n_1} |\Sigma_1^{(j)}|) + \frac{1}{2} \ln((2\pi e)^{n_2} |\Sigma_2^{(j)}|) - \frac{1}{2} \ln((2\pi e)^{n_1 + n_2} |\Sigma^{(j)}|) \tag{29}$$

$$= \frac{1}{2} \ln \left( \frac{|\Sigma_1^{(j)}||\Sigma_2^{(j)}|}{|\Sigma^{(j)}|} \right) + \frac{1}{2} \ln \left( \frac{(2\pi e)^{n_1} (2\pi e)^{n_2}}{(2\pi e)^{n_1 + n_2}} \right) \tag{30}$$

$$= \frac{1}{2} \ln \left( \frac{|\Sigma_1^{(j)}||\Sigma_2^{(j)}|}{|\Sigma^{(j)}|} \right) \tag{31}$$

Summing over all dimensions yields the result. $\qquad\square$

In practice, to ensure that the covariance matrix was invertible, we stayed with a low temporal resolution, from $k = 5, k' = 10$ to $k = 30, k' = 40$. If the covariance matrix was not invertible, we were just removing it from our results. To perform the experiences, we have chosen different Gaussian kernels,

- **AR Kernel (Auto-Regressive):** $K_{\text{AR}}(t_1, t_2) = \sigma^2 \rho^{|t_1 - t_2|}$
- **Matérn 3/2 Kernel:** $K_{\text{M32}}(t_1, t_2) = \sigma^2 \left(1 + \frac{\sqrt{3}\,|t_1 - t_2|}{l}\right) \exp\left(-\frac{\sqrt{3}\,|t_1 - t_2|}{l}\right)$
- **Matérn 5/2 Kernel:** $K_{\text{M52}}(t_1, t_2) = \sigma^2 \left(1 + \frac{\sqrt{5}\,|t_1 - t_2|}{l} + \frac{5\,|t_1 - t_2|^2}{3\,l^2}\right) \exp\left(-\frac{\sqrt{5}\,|t_1 - t_2|}{l}\right)$

- **Squared Exponential Kernel:** $K_{\mathrm{SE}}(t_1, t_2) = \sigma^2 \exp\left(-\frac{|t_1-t_2|^2}{2l^2}\right)$

- **Periodic Kernel:** $K_{\mathrm{per}}(t_1, t_2) = \sigma^2 \exp\left(-\frac{2\sin^2\left(\pi\,\frac{|t_1-t_2|}{p}\right)}{l^2}\right)$

- **Rational Quadratic Kernel:** $K_{\mathrm{RQ}}(t_1, t_2) = \sigma^2 \left(1 + \frac{|t_1-t_2|^2}{2\theta\,l^2}\right)^{-\theta}$

- **Locally Periodic Kernel:** $K_{\mathrm{LP}}(t_1, t_2) = \sigma^2 \exp\left(-\frac{2\sin^2\left(\pi\,\frac{|t_1-t_2|}{p}\right)}{l^2}\right)\exp\left(-\frac{|t_1-t_2|^2}{2d^2}\right)$

For our experiment, we evaluated several variational lower bounds for mutual information estimation : $\hat{\mathbf{I}}_{\mathrm{pred}}{-}\texttt{SMILE}$ [52], $\hat{\mathbf{I}}_{\mathrm{pred}}{-}\texttt{NWJ}$ [39], $\hat{\mathbf{I}}_{\mathrm{pred}}{-}\texttt{InfoNCE}$ [41], $\hat{\mathbf{I}}_{\mathrm{pred}}{-}\texttt{DV}$ [5], and $\hat{\mathbf{I}}_{\mathrm{pred}}{-}\texttt{TUBA}$ [43]. We took as parameters the followings, where in practice, $t_1$ and $t_2$ takes values between $0$ and $k + k\prime - 1$. As we choose $t = k - 1$ as the date of observation of $\{X_t\}_{t-k+1}^{t+k'}$. The table 3 describe the parameters we took for each of the kernels in the experiments.

| Method | $\rho$ | $l$ | **Period** | **Decay** | $\theta$ | $\sigma$ |
|---|---|---|---|---|---|---|
| AR | 0.8 | – | – | – | – | 0.5 |
| Matérn 3/2 | – | 2.0 | – | – | – | 1.0 |
| Matérn 5/2 | – | 2.0 | – | – | – | 1.0 |
| Squared Exponential | – | 2.0 | – | – | – | 1.0 |
| Periodic | – | 3.0 | 2.0 | – | – | 0.5 |
| Rational Quadratic | – | 2.0 | – | – | 1.0 | 1.0 |
| Locally Periodic | – | 1.0 | 4.0 | 10.0 | – | 1.0 |

Table 3: Parameters we choose for each of the kernels.

## B.3  Additional Combinations of $(k, k')$ for Predictive Information Estimation

This section presents complementary estimation results for the predictive information $\mathbf{I}_{\mathrm{pred}}(k, k')$ obtained with various combinations of past and future context lengths $(k, k')$. These results extend the main analysis shown in Figure 1, where $k = 5$ and $k' = 10$ were fixed.

**Case $k = 2$, $k' = 5$:**

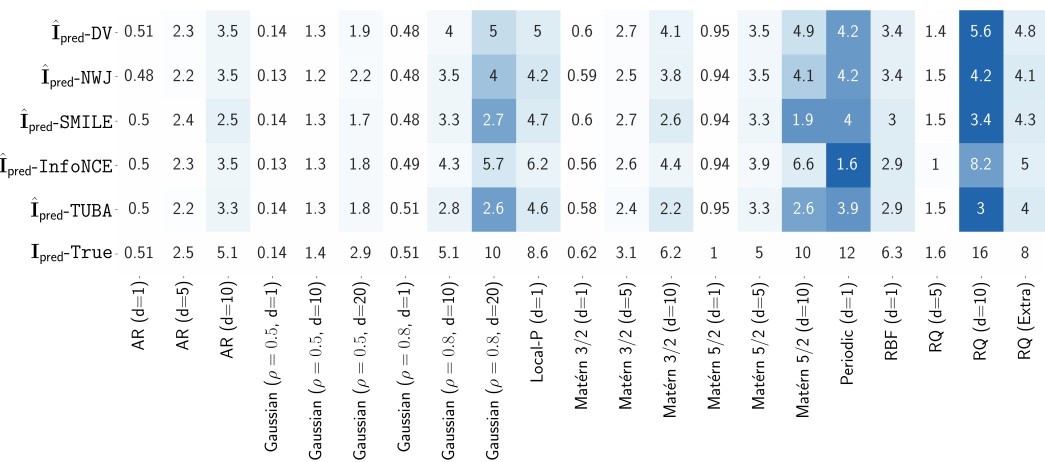

Figure 3: Estimation of $\mathbf{I}_{\mathrm{pred}}(2, 5)$ using various neural-based methods.

**Case $k = 10$, $k' = 20$:**

| | AR (d=1) | AR (d=5) | AR (d=10) | Gaussian ($\rho=0.5$, d=1) | Gaussian ($\rho=0.5$, d=10) | Gaussian ($\rho=0.5$, d=20) | Gaussian ($\rho=0.8$, d=1) | Gaussian ($\rho=0.8$, d=10) | Gaussian ($\rho=0.8$, d=20) | Local-P (d=1) | Matérn 3/2 (d=1) | Matérn 3/2 (d=5) | Matérn 3/2 (d=10) | Matérn 5/2 (d=1) | Matérn 5/2 (d=5) | Matérn 5/2 (d=10) | Periodic (d=1) | RBF (d=1) | RQ (d=5) | RQ (d=10) | RQ (Extra) |
|---|---|---|---|---|---|---|---|---|---|---|---|---|---|---|---|---|---|---|---|---|---|
| $\hat{\mathbf{I}}_{\mathrm{pred}}$-DV | 0.49 | 2.3 | 3.1 | 0.14 | 1.1 | 1.1 | 0.51 | 3.5 | 3.4 | 5 | 0.6 | 2.5 | 4.2 | 0.93 | 3.7 | 4.9 | 4.3 | 3.3 | 1.4 | 6.3 | 4.9 |
| $\hat{\mathbf{I}}_{\mathrm{pred}}$-NWJ | 0.5 | 2.2 | 3.5 | 0.13 | 1.2 | 1.2 | 0.48 | 3.4 | 3.7 | 4.1 | 0.6 | 2.6 | 3.8 | 0.94 | 3.5 | 4.1 | 4.2 | 3.6 | 1.5 | 4.2 | 4.1 |
| $\hat{\mathbf{I}}_{\mathrm{pred}}$-SMILE | 0.49 | 2.3 | 3 | 0.14 | 1 | 1 | 0.49 | 2.9 | 2.5 | 4.6 | 0.6 | 2.5 | 3.1 | 0.95 | 2.5 | 2.2 | 3.7 | 3 | 1.4 | 1.9 | 3.8 |
| $\hat{\mathbf{I}}_{\mathrm{pred}}$-InfoNCE | 0.51 | 2.2 | 3.5 | 0.14 | 0.87 | 0.96 | 0.49 | 3.8 | 3.6 | 4.9 | 0.58 | 2.7 | 4.5 | 0.95 | 3.8 | 6.3 | 4.2 | 3 | 1.2 | 9.5 | 5 |
| $\hat{\mathbf{I}}_{\mathrm{pred}}$-TUBA | 0.5 | 2.3 | 3.3 | 0.14 | 1.1 | 1.2 | 0.49 | 2.7 | 3 | 4.7 | 0.6 | 2.5 | 3 | 0.96 | 3.5 | 3.3 | 3.5 | 3.1 | 1.5 | 3 | 3.7 |
| $\mathbf{I}_{\mathrm{pred}}$-True | 0.51 | 2.5 | 5.1 | 0.14 | 1.4 | 2.9 | 0.51 | 5.1 | 10 | 8.6 | 0.62 | 3.1 | 6.2 | 1 | 5 | 10 | 12 | 6.3 | 1.6 | 16 | 8 |

Figure 4: Estimation of $\mathbf{I}_{\mathrm{pred}}(10, 20)$ using various neural-based methods.

## B.4 Other simple process

Consider the process $\{z_t\}_{t=0}^{N-1} \subset \mathbb{R}^d$ defined as follows:

$$z_0 \sim \mathcal{N}(0, I_d), \tag{32}$$

$$z_t = \rho\, z_{t-1} + \sqrt{1-\rho^2}\, \varepsilon_t, \quad t \geq 1, \tag{33}$$

where $\rho \in (-1, 1)$ is the correlation coefficient, $I_d$ is the $d \times d$ identity matrix, and $\varepsilon_t \sim \mathcal{N}(0, I_d)$ are i.i.d. random vectors.

This defines a Markov (AR(1)) Gaussian process in which each state depends solely on the immediate predecessor. If the total time length is given by $N = T_{\mathrm{past}} + T_{\mathrm{fut}}$, we partition the sequence into:

$$X_{\mathrm{past}} = \{z_0, z_1, \ldots, z_{T_{\mathrm{past}}-1}\},$$
$$X_{\mathrm{fut}} = \{z_{T_{\mathrm{past}}}, \ldots, z_{N-1}\}.$$

Then we compute the theorical value of $\mathbf{I}_{\mathrm{pred}}$ thanks to 4.2.

## B.5 Estimator comparison for $T_{past} = 30$, $T_{future} = 40$

Below are the estimated values of $\mathbf{I}_{\mathrm{pred}}$ across different kernel types, critic architectures, and $\rho$ values. The training parameters are as follows: a batch size of 70, 10,000 iterations, and a learning rate of $5 \times 10^{-4}$.

| dim | theoretical_mi | rho | SeparableCritic | ConcatCritic | SequentialCritic | EvoRate | kernel_type |
|---|---|---|---|---|---|---|---|
| 1 | 0.14 | 0.50 | 0.14 | 0.14 | 0.01 | **0.14** | |
| 1 | 0.51 | 0.80 | 0.48 | **0.51** | 0.28 | 0.48 | |
| 1 | 0.83 | 0.90 | 0.77 | **0.82** | 0.44 | 0.79 | |
| 5 | 0.72 | 0.50 | 0.63 | **0.67** | -0.07 | 0.65 | |
| 5 | 2.55 | 0.80 | 2.16 | **2.41** | 1.37 | 2.13 | |
| 5 | 4.15 | 0.90 | 3.17 | **3.75** | 2.15 | 3.34 | |
| 20 | 2.88 | 0.50 | 0.60 | **1.45** | -1.79 | 0.60 | |
| 20 | 10.22 | 0.80 | 3.21 | **6.07** | 5.86 | 3.16 | |
| 20 | 16.61 | 0.90 | 4.97 | **9.28** | 8.87 | 4.95 | |
| 100 | 14.38 | 0.50 | 0.41 | 0.78 | **5.14** | 0.51 | |
| 100 | 51.08 | 0.80 | 2.13 | 3.76 | **45.64** | 2.46 | |
| 100 | 83.04 | 0.90 | 4.40 | 5.46 | **63.96** | 4.58 | |
| 1 | 0.51 | | 0.48 | **0.50** | 0.28 | 0.47 | AR |
| 1 | 0.62 | | 0.58 | **0.59** | 0.23 | 0.57 | matern32 |
| 1 | | | -6.83 | 7.09 | 0.58 | 1.39 | periodic |
| 5 | 2.55 | | 1.83 | **2.39** | 1.10 | 2.22 | AR |
| 5 | 3.09 | | 2.52 | **2.75** | 0.97 | 2.62 | matern32 |
| 5 | | | 7.12 | 28.55 | 2.83 | 3.93 | periodic |
| 20 | 10.22 | | 2.12 | **4.92** | 4.08 | 2.11 | AR |
| 20 | 12.35 | | 2.33 | 5.40 | **5.45** | 3.23 | matern32 |
| 20 | | | 13.34 | 31.06 | 11.61 | 4.96 | periodic |
| 100 | 51.08 | | 2.40 | 3.63 | **20.77** | 1.87 | AR |
| 100 | 61.74 | | 2.07 | 3.06 | **44.52** | 2.37 | matern32 |
| 100 | | | 9.47 | 26.18 | 81.52 | 7.40 | periodic |

Table 4: Comparison of predictive information estimators across different kernel types and process dimensions. Values rounded to two decimals.

## C  Experiment

**Note on the Use of Nats**

**Units in nats.** Throughout our experiments, all information-theoretic results and measures (e.g., mutual information and differential entropy) are reported in *nats* rather than in bits. In information theory, the choice of base for the logarithm determines the unit: base-$e$ (the natural logarithm) yields nats, while base-2 yields bits. We prefer natural logarithms because they often simplify analytical expressions in both theory and implementation (e.g., when computing the log-likelihood in many machine learning frameworks). However, it is straightforward to convert from nats to bits by noting

$$1 \text{ nat } = \frac{1}{\ln(2)} \text{ bits.}$$

Due to this simple relationship, one can easily switch to bits by scaling the values by $1/\ln(2)$ if desired.

**Estimation of $\tilde{\Lambda}(k)$**

To estimate the learning curve $\tilde{\Lambda}(k) = l(k) - l_0$, we evaluate the conditional entropy rate $l(k)$ from the data and analytically compute the theoretical baseline $l_0$. From the probabilistic formulation in Equation (9), we can directly access the data distribution, which allows us to compute the entropy as follows:

**Estimation of $l(k)$.** We estimate the conditional entropy $l(k) = H(X_t \mid X_{t-k}, \ldots, X_{t-1})$ of a multivariate time series using ridge regression. After fitting a linear model to predict $X_t$ from its past, we compute the residuals and estimate their empirical covariance matrix $\Sigma$. Assuming the residuals are approximately Gaussian, the conditional entropy is estimated using:

$$l(k) \approx \frac{1}{2} \ln \left( (2\pi e)^d \cdot |\Sigma| \right),$$

where $d$ is the dimension of the observed vectors.

**Estimation of $l_0$.**

According to Proposition 4.2, we have $l_0 = l(p)$ for an autoregressive process of order $p$. Furthermore, from Equation 9, $X_t \mid X_{t-1}, \ldots, X_{t-p}$ follows a normal distribution with

$$\mu = \frac{\rho}{p} \sum_{j=t-p}^{t-1} X_j \quad \text{and} \quad \sigma^2 = 1 - \rho^2.$$

Since $l(p) = H(X_t \mid X_{t-1}, \ldots, X_{t-p})$, we can compute it directly using the above formula for the conditional entropy of a multivariate normal distribution.

## C.1    Ising Spin Sequences

**Training of the MLP and LSTM models.** We train both a multilayer perceptron (MLP) and a long short-term memory (LSTM) model on spin chain data generated using the procedure described in Section 5.3. Each model receives a context window of length $k$ and is trained to predict the next binary symbol in the sequence.

The MLP model consists of two hidden layers with ReLU activations, mapping the input vector of length $k$ to a softmax output over two classes. The LSTM model, on the other hand, processes the input sequence as a series of scalar values through an LSTM layer followed by a fully connected output layer.

Training is performed using the Adam optimizer with a learning rate of $10^{-3}$ and a batch size of 128. For each value of $k$, training proceeds for up to 1000 epochs, using early stopping with a patience of 10 epochs based on validation loss. To standardize comparisons across models, we fix the number of batches per epoch and apply the same evaluation procedure to all architectures. The dataset is split into 80% training and 20% validation sets, and results are averaged across runs to account for variance.

**Model architectures:**

- **MLP**: 2 hidden layers (64, then 32 neurons), ReLU activations, output layer with 2 units (softmax).
- **LSTM**: Single-layer LSTM (32 hidden units), fully connected layer mapping to 2 output classes.

**Training setup:** Adam optimizer (lr=$10^{-3}$), batch size=128, max epochs=1000, early stopping (patience=10), 50 batches per epoch, 80/20 train-validation split.

Below are the results obtained for different block sizes *M*: (1) 10,000, (2) 100,000, (3) 1,000,000, and (4) 10,000,000.

| $k$ | $\hat{\mathcal{R}}_{\mathrm{LSTM}}^k(Q)$ | $\hat{\mathcal{R}}_{\mathrm{MLP}}^k(Q)$ | $\hat{\Lambda}(k)$ | $\hat{\mathcal{R}}_{\mathrm{LSTM}}^k(Q) - \hat{\Lambda}(k)$ | $\hat{\mathcal{R}}_{\mathrm{MLP}}^k(Q) - \hat{\Lambda}(k)$ | EvoRate$(k)$ |
|---|---|---|---|---|---|---|
| 1 | 0.6932 | 0.6927 | 0.3208 ± 0.0043 | **0.3724 ± 0.0043** | **0.3719 ± 0.0043** | 0.0005 ± 0.0012 |
| 2 | 0.6220 | 0.6200 | 0.2058 ± 0.0023 | 0.4162 ± 0.0023 | 0.4142 ± 0.0023 | 0.1171 ± 0.0080 |
| 3 | 0.5955 | 0.5571 | 0.1503 ± 0.0031 | 0.4452 ± 0.0031 | 0.4068 ± 0.0031 | 0.1731 ± 0.0105 |
| 4 | 0.5611 | 0.5560 | 0.1177 ± 0.0034 | 0.4434 ± 0.0034 | 0.4383 ± 0.0034 | 0.2059 ± 0.0115 |
| 5 | 0.5394 | 0.5429 | 0.0962 ± 0.0035 | 0.4432 ± 0.0035 | 0.4467 ± 0.0035 | 0.2274 ± 0.0120 |
| 6 | 0.5347 | 0.5343 | 0.0810 ± 0.0036 | 0.4537 ± 0.0036 | 0.4533 ± 0.0036 | 0.2427 ± 0.0122 |
| 7 | 0.5167 | 0.5199 | 0.0696 ± 0.0035 | 0.4471 ± 0.0035 | 0.4503 ± 0.0035 | 0.2541 ± 0.0124 |
| 8 | 0.5105 | 0.5088 | 0.0607 ± 0.0035 | 0.4498 ± 0.0035 | 0.4481 ± 0.0035 | 0.2629 ± 0.0125 |
| 9 | 0.5036 | 0.5092 | 0.0537 ± 0.0034 | 0.4499 ± 0.0034 | 0.4555 ± 0.0034 | 0.2700 ± 0.0126 |
| 10 | 0.5079 | 0.5064 | 0.0479 ± 0.0034 | 0.4600 ± 0.0034 | 0.4585 ± 0.0034 | 0.2758 ± 0.0126 |
| 11 | 0.5077 | 0.5232 | 0.0430 ± 0.0034 | 0.4647 ± 0.0034 | 0.4802 ± 0.0034 | 0.2807 ± 0.0127 |
| 12 | 0.5181 | 0.4997 | 0.0389 ± 0.0034 | 0.4792 ± 0.0034 | 0.4608 ± 0.0034 | 0.2850 ± 0.0127 |
| 13 | 0.4965 | 0.5012 | 0.0354 ± 0.0033 | 0.4611 ± 0.0033 | 0.4658 ± 0.0033 | 0.2887 ± 0.0128 |
| 14 | 0.4989 | 0.4992 | 0.0324 ± 0.0033 | 0.4665 ± 0.0033 | 0.4668 ± 0.0033 | 0.2924 ± 0.0128 |
| 15 | 0.4864 | 0.5000 | 0.0297 ± 0.0032 | 0.4567 ± 0.0032 | 0.4703 ± 0.0032 | 0.2962 ± 0.0128 |
| 16 | 0.4949 | 0.5062 | 0.0273 ± 0.0032 | 0.4676 ± 0.0032 | 0.4789 ± 0.0032 | 0.3010 ± 0.0128 |
| 17 | 0.4972 | 0.4968 | 0.0252 ± 0.0032 | 0.4720 ± 0.0032 | 0.4716 ± 0.0032 | 0.3080 ± 0.0128 |
| 18 | 0.4885 | **0.4872** | 0.0232 ± 0.0033 | 0.4653 ± 0.0033 | 0.4640 ± 0.0033 | 0.3202 ± 0.0128 |
| 19 | **0.4853** | 0.5041 | 0.0213 ± 0.0034 | 0.4640 ± 0.0034 | 0.4828 ± 0.0034 | 0.3452 ± 0.0129 |
| **Minima** | **0.4853** | **0.4872** | – | $\hat{\mathcal{R}}^\infty(Q^*) = 0.3724 \pm 0.0043$ | $\hat{\mathcal{R}}^\infty(Q^*) = 0.3719 \pm 0.0043$ | – |

Table 5: Estimated risks and $\Lambda(k)$ values for different history lengths $k$ ($M = 10{,}000$).

| Order $k$ | $\hat{\mathcal{R}}^k_{\text{LSTM}}(Q)$ | $\hat{\mathcal{R}}^k_{\text{MLP}}(Q)$ | $\hat{\Lambda}(k)$ | $\hat{\mathcal{R}}^k_{\text{LSTM}}(Q) - \hat{\Lambda}(k)$ | $\hat{\mathcal{R}}^k_{\text{MLP}}(Q) - \hat{\Lambda}(k)$ | EvoRate($k$) |
|---|---|---|---|---|---|---|
| 1 | 0.6897 | 0.6933 | 0.3213 ± 0.0153 | **0.3684 ± 0.0153** | **0.3720 ± 0.0153** | 0.0056 ± 0.0030 |
| 2 | 0.5789 | 0.6070 | 0.2061 ± 0.0058 | 0.3728 ± 0.0058 | 0.4009 ± 0.0058 | 0.1261 ± 0.0208 |
| 3 | 0.6036 | 0.5745 | 0.1506 ± 0.0071 | 0.4530 ± 0.0071 | 0.4239 ± 0.0071 | 0.1832 ± 0.0262 |
| 4 | 0.5497 | 0.5615 | 0.1179 ± 0.0081 | 0.4318 ± 0.0081 | 0.4436 ± 0.0081 | 0.2163 ± 0.0282 |
| 5 | 0.5146 | 0.5383 | 0.0964 ± 0.0085 | 0.4182 ± 0.0085 | 0.4419 ± 0.0085 | 0.2379 ± 0.0290 |
| 6 | 0.5226 | 0.5537 | 0.0811 ± 0.0087 | 0.4415 ± 0.0087 | 0.4726 ± 0.0087 | 0.2531 ± 0.0294 |
| 7 | 0.5311 | 0.5409 | 0.0697 ± 0.0088 | 0.4614 ± 0.0088 | 0.4712 ± 0.0088 | 0.2645 ± 0.0296 |
| 8 | 0.5124 | 0.5531 | 0.0608 ± 0.0088 | 0.4516 ± 0.0088 | 0.4923 ± 0.0088 | 0.2733 ± 0.0296 |
| 9 | 0.5236 | 0.5331 | 0.0537 ± 0.0088 | 0.4699 ± 0.0088 | 0.4794 ± 0.0088 | 0.2803 ± 0.0297 |
| 10 | 0.4901 | 0.5186 | 0.0479 ± 0.0088 | 0.4422 ± 0.0088 | 0.4707 ± 0.0088 | 0.2861 ± 0.0297 |
| 11 | **0.4679** | 0.5119 | 0.0430 ± 0.0088 | 0.4249 ± 0.0088 | 0.4689 ± 0.0088 | 0.2910 ± 0.0296 |
| 12 | 0.4975 | 0.5273 | 0.0389 ± 0.0088 | 0.4586 ± 0.0088 | 0.4884 ± 0.0088 | 0.2952 ± 0.0296 |
| 13 | 0.4692 | 0.4736 | 0.0353 ± 0.0088 | 0.4339 ± 0.0088 | 0.4383 ± 0.0088 | 0.2990 ± 0.0296 |
| 14 | 0.5152 | 0.5259 | 0.0323 ± 0.0088 | 0.4829 ± 0.0088 | 0.4936 ± 0.0088 | 0.3026 ± 0.0296 |
| 15 | 0.4971 | 0.4733 | 0.0296 ± 0.0087 | 0.4675 ± 0.0087 | 0.4437 ± 0.0087 | 0.3064 ± 0.0296 |
| 16 | 0.4714 | 0.5184 | 0.0272 ± 0.0087 | 0.4442 ± 0.0087 | 0.4912 ± 0.0087 | 0.3112 ± 0.0296 |
| 17 | 0.4996 | 0.5278 | 0.0250 ± 0.0087 | 0.4746 ± 0.0087 | 0.5028 ± 0.0087 | 0.3182 ± 0.0295 |
| 18 | 0.5037 | 0.4748 | 0.0230 ± 0.0089 | 0.4807 ± 0.0089 | 0.4518 ± 0.0089 | 0.3305 ± 0.0297 |
| 19 | 0.5189 | **0.4668** | 0.0211 ± 0.0092 | 0.4978 ± 0.0092 | 0.4457 ± 0.0092 | 0.3556 ± 0.0299 |
| **Minima** | **0.4679** | **0.4668** | – | $\hat{\mathcal{R}}^\infty(Q^*) = 0.3684 \pm 0.0153$ | $\hat{\mathcal{R}}^\infty(Q^*) = 0.3720 \pm 0.0153$ | – |

Table 6: Estimated risks and $\Lambda(k)$ values for different history lengths $k$ ($M = 100{,}000$).

| Order $k$ | $\hat{\mathcal{R}}^k_{\text{LSTM}}(Q)$ | $\hat{\mathcal{R}}^k_{\text{MLP}}(Q)$ | $\hat{\Lambda}(k)$ | $\hat{\mathcal{R}}^k_{\text{LSTM}}(Q) - \hat{\Lambda}(k)$ | $\hat{\mathcal{R}}^k_{\text{MLP}}(Q) - \hat{\Lambda}(k)$ | EvoRate($k$) |
|---|---|---|---|---|---|---|
| 1 | 0.6910 | 0.6808 | 0.3235 ± 0.0718 | 0.3675 ± 0.0718 | 0.3573 ± 0.0718 | 0.0009 ± 0.0320 |
| 2 | 0.5641 | 0.6290 | 0.2069 ± 0.0188 | 0.3572 ± 0.0188 | 0.4221 ± 0.0188 | 0.1450 ± 0.0465 |
| 3 | 0.6157 | 0.5502 | 0.1510 ± 0.0314 | 0.4647 ± 0.0314 | 0.3992 ± 0.0314 | 0.2090 ± 0.0554 |
| 4 | 0.5913 | 0.5847 | 0.1181 ± 0.0376 | 0.4732 ± 0.0376 | 0.4666 ± 0.0376 | 0.2460 ± 0.0588 |
| 5 | 0.5621 | 0.5145 | 0.0965 ± 0.0395 | 0.4656 ± 0.0395 | 0.4180 ± 0.0395 | 0.2706 ± 0.0603 |
| 6 | 0.5087 | 0.6114 | 0.0811 ± 0.0402 | 0.4276 ± 0.0402 | 0.5303 ± 0.0402 | 0.2883 ± 0.0609 |
| 7 | 0.4705 | 0.4423 | 0.0697 ± 0.0401 | 0.4008 ± 0.0401 | 0.3726 ± 0.0401 | 0.3016 ± 0.0613 |
| 8 | 0.5378 | 0.5469 | 0.0607 ± 0.0398 | 0.4771 ± 0.0398 | 0.4862 ± 0.0398 | 0.3120 ± 0.0614 |
| 9 | 0.5195 | 0.5008 | 0.0536 ± 0.0394 | 0.4659 ± 0.0394 | 0.4472 ± 0.0394 | 0.3202 ± 0.0615 |
| 10 | 0.4480 | 0.4690 | 0.0478 ± 0.0390 | 0.4002 ± 0.0390 | 0.4212 ± 0.0390 | 0.3269 ± 0.0615 |
| 11 | 0.5612 | 0.5034 | 0.0429 ± 0.0386 | 0.5183 ± 0.0386 | 0.4605 ± 0.0386 | 0.3326 ± 0.0615 |
| 12 | 0.5172 | 0.4650 | 0.0388 ± 0.0383 | 0.4784 ± 0.0383 | 0.4262 ± 0.0383 | 0.3373 ± 0.0615 |
| 13 | 0.5039 | 0.5169 | 0.0352 ± 0.0379 | 0.4687 ± 0.0379 | 0.4817 ± 0.0379 | 0.3416 ± 0.0615 |
| 14 | 0.5588 | 0.4728 | 0.0321 ± 0.0376 | 0.5267 ± 0.0376 | 0.4407 ± 0.0376 | 0.3456 ± 0.0615 |
| 15 | 0.4760 | 0.4402 | 0.0294 ± 0.0374 | 0.4466 ± 0.0374 | 0.4108 ± 0.0374 | 0.3498 ± 0.0614 |
| 16 | 0.5425 | **0.3660** | 0.0270 ± 0.0373 | 0.5155 ± 0.0373 | **0.3390 ± 0.0373** | 0.3548 ± 0.0614 |
| 17 | 0.4003 | 0.4738 | 0.0249 ± 0.0370 | 0.3754 ± 0.0370 | 0.4489 ± 0.0370 | 0.3619 ± 0.0616 |
| 18 | **0.3798** | 0.4653 | 0.0229 ± 0.0359 | **0.3569 ± 0.0359** | 0.4424 ± 0.0359 | 0.3741 ± 0.0618 |
| 19 | 0.5082 | 0.5629 | 0.0210 ± 0.0315 | 0.4872 ± 0.0315 | 0.5419 ± 0.0315 | 0.3989 ± 0.0618 |
| **Minima** | **0.3798** | **0.3660** | – | $\hat{\mathcal{R}}^\infty(Q^*) = 0.3569 \pm 0.0359$ | $\hat{\mathcal{R}}^\infty(Q^*) = 0.3390 \pm 0.0373$ | – |

Table 7: Estimated risks and $\Lambda(k)$ values for different history lengths $k$ ($M = 1{,}000{,}000$).

| Order $k$ | $\hat{\mathcal{R}}^k_{\text{LSTM}}(Q)$ | $\hat{\mathcal{R}}^k_{\text{MLP}}(Q)$ | $\hat{\Lambda}(k)$ | $\hat{\mathcal{R}}^k_{\text{LSTM}}(Q) - \hat{\Lambda}(k)$ | $\hat{\mathcal{R}}^k_{\text{MLP}}(Q) - \hat{\Lambda}(k)$ | EvoRate(k) |
|---|---|---|---|---|---|---|
| 1 | 0.6709 | 0.3551 | 0.0037 ± 0.0929 | 0.6672 ± 0.0929 | 0.3514 ± 0.0929 | 0.4760 ± 0.2277 |
| 2 | 0.5209 | 0.2713 | 0.0037 ± 0.0067 | 0.5172 ± 0.0067 | 0.2676 ± 0.0067 | 0.4760 ± 0.2277 |
| 3 | 0.1961 | 0.6698 | 0.0037 ± 0.0086 | 0.1924 ± 0.0086 | 0.6661 ± 0.0086 | 0.4760 ± 0.2277 |
| 4 | 0.5440 | 0.6037 | 0.0037 ± 0.0075 | 0.5403 ± 0.0075 | 0.6000 ± 0.0075 | 0.4760 ± 0.2277 |
| 5 | 0.2271 | 0.1029 | 0.0037 ± 0.0108 | 0.2234 ± 0.0108 | 0.0992 ± 0.0108 | 0.4760 ± 0.2277 |
| 6 | 0.2312 | 0.4198 | 0.0037 ± 0.0101 | 0.2275 ± 0.0101 | 0.4161 ± 0.0101 | 0.4760 ± 0.2277 |
| 7 | 0.6919 | 0.6896 | 0.0037 ± 0.0139 | 0.6882 ± 0.0139 | 0.6859 ± 0.0139 | 0.4760 ± 0.2277 |
| 8 | 0.6824 | 0.5694 | 0.0037 ± 0.0143 | 0.6787 ± 0.0143 | 0.5657 ± 0.0143 | 0.4760 ± 0.2277 |
| 9 | 0.6913 | **0.0903** | 0.0036 ± 0.0061 | 0.6877 ± 0.0061 | **0.0867 ± 0.0061** | 0.4760 ± 0.2277 |
| 10 | **0.0733** | 0.1276 | 0.0036 ± 0.0206 | **0.0697 ± 0.0206** | 0.1240 ± 0.0206 | 0.4760 ± 0.2277 |
| 11 | 0.5529 | 0.6932 | 0.0035 ± 0.0067 | 0.5494 ± 0.0067 | 0.6897 ± 0.0067 | 0.4761 ± 0.2277 |
| 12 | 0.5981 | 0.6909 | 0.0034 ± 0.0098 | 0.5947 ± 0.0098 | 0.6875 ± 0.0098 | 0.4762 ± 0.2277 |
| 13 | 0.5111 | 0.6873 | 0.0031 ± 0.0079 | 0.5080 ± 0.0079 | 0.6842 ± 0.0079 | 0.4764 ± 0.2276 |
| 14 | 0.4116 | 0.5636 | 0.0025 ± 0.0063 | 0.4091 ± 0.0063 | 0.5611 ± 0.0063 | 0.4767 ± 0.2275 |
| 15 | 0.6620 | 0.5387 | 0.0013 ± 0.0054 | 0.6607 ± 0.0054 | 0.5374 ± 0.0054 | 0.4772 ± 0.2271 |
| 16 | 0.2525 | 0.6352 | -0.0010 ± 0.0099 | 0.2535 ± 0.0099 | 0.6362 ± 0.0099 | 0.4779 ± 0.2263 |
| 17 | 0.3450 | 0.5577 | -0.0059 ± 0.0138 | 0.3509 ± 0.0138 | 0.5636 ± 0.0138 | 0.4788 ± 0.2246 |
| 18 | 0.6929 | 0.6884 | -0.0156 ± 0.0142 | 0.7085 ± 0.0142 | 0.7040 ± 0.0142 | 0.4801 ± 0.2209 |
| 19 | 0.2151 | 0.2619 | -0.0355 ± 0.0050 | 0.2506 ± 0.0050 | 0.2974 ± 0.0050 | 0.4819 ± 0.2128 |
| **Minima** | **0.0733** | **0.0903** | – | $\hat{\mathcal{R}}^\infty(Q^*) = 0.0697 \pm 0.0206$ | $\hat{\mathcal{R}}^\infty(Q^*) = 0.0867 \pm 0.0061$ | – |

Table 8: Estimated risks and $\Lambda(k)$ values for different history lengths $k$ ($M = 10{,}000{,}000$).

# D  Computational Cost

All experiments were conducted on a system equipped with a 14-core CPU, a 20-core integrated GPU, 24 GB of unified memory, and 1 TB of SSD storage.

**Estimating $\mathbf{I}_{\text{pred}}$** The computational complexity of the proposed algorithm is primarily driven by the estimation of mutual information between learned representations. This process incurs a cost of $\mathcal{O}(B^2 d)$ per iteration, where $B$ is the batch size and $d$ is the dimensionality of the representations. The $\mathcal{O}(B^2 d)$ complexity reflects the need to compute pairwise interactions between samples within each batch in a high-dimensional space. The total computational cost per iteration is therefore $\mathcal{O}(B^2 d)$, excluding the contribution of the encoder $g$ and decoder $h$, whose complexity depends on their specific architectures.

**Estimating $\hat{\Lambda}(k)$** Using Corollary 4.4, when $k$ ranges from 1 to $n$, the computational complexity of estimating the learning curve is $\mathcal{O}(n B^2 d)$.

**Estimating $\mathcal{R}^\infty(Q^*)$** The total cost corresponds to training the model $Q$ times over $n$. However, this cost is generally unknown or difficult to quantify directly. Moreover, it must be added to the computational cost required to estimate the learning curve $\hat{\Lambda}(k)$.

# E  Algorithms training procedures

---

**Algorithm 1 $\mathbf{I}_{\text{pred}}$:** Data is sampled in a sequential manner with temporal alignment

---

1: **for each training iteration do**
2:  Sample $\{(x_i, y_i)\}_{i=1}^{B}$ from a long sequence $\{z_t\}_{t=1}^{N}$ such that:

$$x_i = [z_{t_i - k}, \ldots, z_{t_i - 1}], \quad y_i = [z_{t_i}, \ldots, z_{t_i + k' - 1}]$$

3:  Compute critic scores $S_{ij} = f_\theta(x_i, y_j)$
4:  Compute $\mathbf{I}_{\text{pred}}$ using Eq. (3):

$$\mathbf{I}_{\text{pred},i}(k, k') := \frac{1}{B} \sum_{i=1}^{B} f_\theta(x_i, y_i) - \ln \left( \frac{1}{B(B-1)} \sum_{\substack{i,j=1 \\ i \neq j}}^{B} \exp(f_\theta(x_i, y_j)) \right)$$

5:  Update critic parameters $\theta$ by maximizing $\mathbf{I}_{\text{pred}}$
6: **end for**

---

# F  Limitations

Estimating $\mathbf{I}_{\text{pred}}$ remains a challenging task in practice. Since the learning curve is derived directly from this quantity, any noise or bias in the estimation can lead to significant errors in the assessment of the optimal achievable risk. In particular, the tendency of current estimators to systematically underestimate $\mathbf{I}_{\text{pred}}$ can misrepresent the true predictive structure of the data. Moreover, estimation can suffer from variance and occasional instability, especially in high-dimensional settings or when the underlying process has weak temporal dependencies. While developing a definitive and robust estimator of $\mathbf{I}_{\text{pred}}$ is not the main objective of this work, we view it as an important and promising direction for future research in its own right.

Another limitation lies in the fact that our estimator of the minimal achievable risk is model-dependent, in contrast to model-free approaches such as `EvoRate` [55]. We acknowledge that this dependency introduces potential confounds related to model capacity and training variability. However, our framework is designed with a different purpose: while `EvoRate` aims to detect the mere presence of temporal structure, our objective is to quantify how much of this structure is captured by a given model class. The core quantity of interest—the learning curve $\Lambda(k)$—remains model-independent and reflects the intrinsic predictability of the data across context lengths. The model-specific risk can then be seen as an added diagnostic, enabling one to assess whether predictive limitations stem from insufficient model capacity or from intrinsic data constraints.

Importantly, in our experiments, the estimated minimal achievable risk was generally stable across different architectures, which suggests that the diagnostic remains robust despite being evaluated within specific model classes. Nonetheless, we emphasize that mutual information estimation in sequential settings is still a young and evolving research area. We hope that our contribution helps lay the theoretical foundation for more robust methods—potentially by shifting focus from direct mutual information estimation to the more stable and interpretable learning curve $\Lambda(k)$.

Finally, an important future direction will be to evaluate the framework on large-scale real-world sequential tasks—such as those in natural language processing—where validating its usefulness beyond synthetic benchmarks would provide strong evidence of its practical relevance.

