# OpenReview forum: "How Patterns Dictate Learnability in Sequential Data"
_NeurIPS.cc/2025/Conference — NeurIPS 2025 poster_

### Official Review · Reviewer_BK66 · 2025-06-22

**Clarity:** 2
**Significance:** 2
**Originality:** 2
**Rating:** 4
**Confidence:** 2

**Summary:**

This paper proposes a theoretically grounded and general framework for understanding the learnability of sequential data using predictive information—the mutual information between the past and the future. It formalizes the concept of an information-theoretic learning curve (via the universal learning curve, Λ(k)) and connects it to fundamental limits on forecasting performance. Building on prior work like EvoRate, the authors extend the analysis by defining a minimal achievable risk, offering new diagnostic tools for model adequacy. Theoretical contributions are complemented by experiments on synthetic datasets, including autoregressive and Ising processes, validating the proposed framework’s ability to detect learnability bounds and optimal regression orders.

**Questions:**

See above.

**Ethical Concerns:**

["NO or VERY MINOR ethics concerns only"]

**Final Justification:**

While the authors' rebuttal was good, I don't feel confident increasing my original score.

**Limitations:**

See above.

**Quality:**

2

**Strengths And Weaknesses:**

Strengths:
	1.	Strong Theoretical Foundations:
The authors rigorously establish the link between predictive information and learnability, grounding their framework in information theory and learning theory. This includes closed-form results for Markov processes, asymptotic expansions for parametric families, and connections to Bayesian learning theory (e.g., Corollary 4.4 on dimensionality estimation).
	2.	Conceptual Clarity:
The paper provides intuitive examples (e.g., lunch meal prediction) to motivate its central question—whether poor forecasting performance arises from data limitations or model inadequacies.
	3.	Sound Use of Information-Theoretic Tools:
The authors successfully extend EvoRate into a more general predictive information metric that connects to the universal learning curve. This supports robust interpretations of structure in data and connects theory with empirical forecasting tasks.
	4.	Synthetic Experiments:
The experiments validate the theory by showing that the proposed estimator Λ(k) tracks ground-truth model orders and minimum risk accurately in controlled settings.

⸻

Weaknesses:
	1.	Empirical Evaluation Scope:
The experiments are performed entirely on synthetic datasets—autoregressive Gaussian processes and Ising spin chains. While these are useful for validation, they fall short of demonstrating the framework’s practical utility on real-world data (e.g., NLP or financial sequences), which the paper itself acknowledges as a limitation.
	2.	Estimation Stability:
The authors note estimation variance and occasional instability of the Ipred estimators in high dimensions or near low-complexity regimes. This potentially limits the utility of the learning curve in real settings unless estimation is improved.
	3.	Model Dependency of the Estimator:
The proposed estimator of minimal achievable risk is model-dependent, which contrasts with EvoRate’s model-free nature. While this allows diagnostic comparisons, it introduces potential confounds related to model capacity and training variability.
	4.	Interpretability vs. Accessibility:
While the theoretical depth is commendable, some derivations and notations—though well-supported in the appendix—could be inaccessible to readers not well-versed in information theory or statistical learning, potentially limiting broader appeal.

Overall Evaluation:

The paper is clearly written, well-structured, and theoretically sound. It presents a significant conceptual advance over EvoRate by offering a quantitative and theoretically justified estimator of the fundamental limits of learnability. While the theory is solid, I cannot fully assess its novelty in the context of the most recent literature on predictive information estimation and sequential risk bounds. As someone less familiar with the deep technical nuances of the information-theoretic theory presented, I found the framework persuasive, but dense.

The empirical component appears somewhat underwhelming compared to the strength of the theoretical contribution. It may be that synthetic validation is standard in this line of work, but the lack of any real-world application or benchmark comparison makes it hard to assess its broader utility.

---

> ### Author Rebuttal · Authors · 2025-07-28
>
> We are grateful to the reviewer for the valuable feedback and recommendations.
> We provide here a detailed answer to the weaknesses and questions.
>
> **W1 – Real world dataset**
>
> We understand that the empirical part of the paper may appear modest relative to the theoretical contribution, which can make evaluating its practical relevance more challenging.
> However, the primary focus of our work is indeed theoretical, and as you may have seen, the theoretical contributions—particularly the proofs—are already substantial and technically demanding.
>
> We intentionally chose not to simultaneously tackle the separate and equally challenging problem of experimentally estimating the learning curve and the minimal achievable risk. Indeed, estimating the learning curve is itself a complex theoretical question that warrants dedicated treatment. Addressing it rigorously would require an entirely separate line of work, and we felt that attempting to incorporate this into the current paper would dilute the clarity and focus of our main contributions. For instance, [1] is a full research article dedicated solely to estimating mutual information in sequential settings. Hence, we chose to first address the problem from a theoretical perspective, and plan to tackle the estimation of the learning curve and thus minimal risk in a follow-up article.
>
> That said, we did include experiments on synthetic data, where the estimation of $\mathbf{I}_{\text{pred}}$ is not an issue, since closed-form expressions are available. These experiments serve an illustrative purpose, allowing us to demonstrate the theoretical framework in controlled settings.
>
> Nevertheless, we fully acknowledge the reviewers' concern regarding the lack of real-world evaluations. In response, we applied our framework to the Exchange Rate dataset from GluonTS [2], and show how it can be used on a real world dataset. This dataset contains daily exchange rates (1990–2016) for eight major currencies (AUD, GBP, CAD, CHF, CNY, JPY, NZD, SGD). In this case, we show that this dataset is more limited by data itself rather than by model complexity.
>
> We first computed the MSE for four models: a naive baseline (which predicts the next value as the previous one), a simple feedforward neural network, DeepAR, and the Temporal Fusion Transformer (TFT). The differences in MSE across models are relatively minor. Even the naive model performs comparably to more sophisticated ones, reinforcing the idea that additional architectural complexity does not necessarily translate into meaningful performance gains.
>
> To obtain the  minimal achievable risk $\hat{\mathcal{R}}^{\infty}(Q^{*}) = \min_{1 \leq k \leq M} \hat{\mathcal{R}}^{k}(Q_k) - \hat{\Lambda}(k)$, we estimated the learning curve $\Lambda(k)$.  While our estimator is admittedly noisy and unstable, the results (shown below) suggest that there is little left to learn in this dataset. Indeed, the actual MSE values lie close to this estimated minimal risk, suggesting that the performance ceiling is due less to model limitations than to the intrinsic unpredictability of the data.
>
> This aligns with the characteristics of the dataset. Unlike many other financial assets, FX rates are driven by the relative economic conditions, monetary policies, and geopolitical factors of two countries, adding significant complexity to their dynamics. This market is also extremely liquid and highly competitive, so any patterns or inefficiencies are quickly detected and eliminated by market participants. These factors contribute to the inherent unpredictability of the dataset.
>
> **Table: Comparison between actual MSE and estimated minimal achievable risk for each model**
>
> | Model| MSE |$\hat{R}^{\infty}(Q^*) $|
> |---------------------|------------------------|----------------------------------------------|
> | Naive  | 0.000030 | 0.000021 ± 0.000007 |
> | DeepAR | 0.000018 ± 0.000004| 0.000013 ± 0.000009|
> | Simple FeedForward  | 0.000021 ± 0.000002    | 0.000015 ± 0.000008|
> | TFT | 0.000016 ± 0.000002| 0.000013 ± 0.000011|
>
> ---
> **W2 – Estimation Stability**
>
> Indeed, as noted above, estimating predictive information in practice is a difficult task—particularly in high-dimensional settings or when the underlying data has weak temporal structure. Our main objective in this work, however, is not to develop a definitive estimator of $\mathbf{I}_{\text{pred}}$, but to provide a theoretical characterization of the limits of learnability from an information-theoretic perspective. We believe that a rigorous and robust estimation procedure for the predictive information is an important and challenging problem in its own right, and we see it as a promising direction for future work.
>
> It is also worth emphasizing that estimating mutual information in sequential settings is a relatively recent and evolving research area—[1] , for instance, was one of the first works to directly address this. While our current estimator is admittedly imperfect, it already yields valuable insights in real-world scenarios, as demonstrated in our experiments on both synthetic and real world datasets.
> Moreover, our theoretical results suggest new avenues for improving estimation techniques. For example, instead of estimating $\mathbf{I}_{\text{pred}}$ directly, one might focus on estimating the learning curve $\Lambda(k)$, which could offer a more stable and interpretable alternative. We view our contribution as laying the groundwork for such future developments.
>
> ---
>
> **W3 – Model Dependency of the Estimator**
>
> We thank the reviewer for this important point. It is true that our estimator of the minimal achievable risk is model-dependent, in contrast to $\texttt{EvoRate}$ 's model-free approach in [1]. However, we believe that this dependency is not necessarily a limitation, but rather reflects the different purpose of our framework. While EvoRate focuses on detecting the presence of temporal structure, our aim is to quantify how much of that structure is actually being captured by a given model class.
>
> The core quantity of interest in our framework—the learning curve $ \Lambda(k) $—is itself model-independent and provides richer information than EvoRate, as it reflects the data’s intrinsic predictability across context lengths. The connection we draw to model-specific risk should be seen as an added benefit: it enables one to assess the current state of modeling capabilities on a given dataset, and to determine whether improvements are more likely to come from better models or whether the data has already been exhaustively exploited.
>
> Importantly, in our experiments, we observed that the estimated minimal achievable risk is generally stable across different architectures. This suggests that our diagnostic remains robust despite being evaluated within specific model classes.
>
> ---
>
> **W4 – Interpretability vs. Accessibility**
>
> We appreciate the reviewer’s concern regarding the accessibility of the theoretical content and recognize that the paper is conceptually dense. To address this, we deliberately included intuitive illustrations—such as the lunch meal prediction scenario in the introduction—to ground the theoretical framework in concrete terms. We also designed empirical experiments, including one on a real-world dataset, to directly support and illustrate our theoretical claims. We began with synthetic data precisely because it offers controlled settings where expected behaviors are known, making the theory more digestible for the reader. Additionally, we will revise and clarify some derivations and reinforce the connection between theory and experiments in the final version of the article to further improve interpretability.
>
> ---
>
> We hope we have adequately addressed the reviewer’s questions and concerns regarding this paper.
>
>
> **References**
>
> [1] Zeng, Q., Huang, L.-K., Chen, Q., Ling, C. X., & Wang, B. (2025). *Towards Understanding Evolving Patterns in Sequential Data*. Advances in Neural Information Processing Systems.
>
> [2] Alexandrov, A., et al. (2019). *GluonTS: Probabilistic Time Series Modeling in Python*. arXiv preprint arXiv:1906.05264.

---

> > ### Comment · Reviewer_BK66 · 2025-08-07
> >
> > I would like to thank the authors for their rebuttal and I will keep my score.

---

### Official Review · Reviewer_sHAW · 2025-07-01

**Clarity:** 3
**Significance:** 3
**Originality:** 3
**Rating:** 5
**Confidence:** 3

**Summary:**

The paper addresses two questions:
1. What are the minimum achievable risks for any predictor intending to model sequential data?
2. Can the error be attributed to predictor limitation or unpredictability of the sequential data?

In the context of stationary processes, the authors consider the mutual information between the past and the future with increasing window size to derive lower bounds for the minimal achievable risks and highlight patterns' impact on learnability. Theoretical results have been highlighted with empirical experiments on synthetic

**Questions:**

- Regarding the additional remarks on Result Interpretability, would it be possible to illustrate corollary 4.4 with a plot showing the asymptotic consistency of the dimension estimator?
- Link to the first question: The window size is closely linked to the parameter space dimension for the autoregressive model. Assuming a large window size, would it be possible to estimate the window size from smaller windows?

**Ethical Concerns:**

["NO or VERY MINOR ethics concerns only"]

**Final Justification:**

I thank the authors for their thorough and clear responses. They have addressed my concerns and provided additional experiments. The paper is easy to follow, and the inclusion of an additional real-world experiment further clarifies the practical usability of the proposed framework. I maintain my score in support of this paper.

**Limitations:**

Limitations are raised in my questions and weaknesses.

**Quality:**

3

**Strengths And Weaknesses:**

Strengths:
- The paper is well-motivated and written. The theoretical framework is well described, and several intuitions are provided, giving a good understanding of the framework's usability.
- Applying the framework to the case of Markovian and parametric processes seems promising, as the estimation of the proper window length or parameter space remains challenging in practice.
- Appendices are well-detailed, complementing both the theoretical section with proofs and the experimental section with detailed results and implementation, easing reproducibility.

Weaknesses:
- While not central to this paper, an illustration of the framework on real-world data would benefit the overall comprehension and paper motivation.
- The paper could also benefit from a summary answering the two introductory questions with the framework as partially done in the theoretical and experimental parts.

---

> ### Author Rebuttal · Authors · 2025-07-29
>
> We thank the reviewer for their thoughtful and constructive feedback. We appreciate the positive assessment of our contributions, as well as the valuable questions and concerns raised. We address them point by point below.
>
> **W1 – Illustration on real-world dataset**
>
> We acknowledge that our work could benefit from an application on a real world dataset.  But the primary focus of our work is theoretical, and as you may have seen, the theoretical contributions—particularly the proofs—are already substantial and technically demanding.
>
> We intentionally chose not to simultaneously tackle the separate and equally challenging problem of experimentally estimating the learning curve and the minimal achievable risk. Indeed, estimating the learning curve is itself a complex theoretical question that warrants dedicated treatment. Addressing it rigorously would require an entirely separate line of work, and we felt that attempting to incorporate this into the current paper would dilute the clarity and focus of our main contributions.
>
> Nevertheless, we agree with the reviewer that a real world application could benefit the overall comprehension of this article. Thus, we applied our estimator of $\mathbf{I}_{\text{pred}}$ to the Exchange Rate dataset from GluonTS [1], in order to assess how much predictive information remains in that data. This dataset contains daily exchange rates (1990–2016) for eight major currencies (AUD, GBP, CAD, CHF, CNY, JPY, NZD, SGD).
>
> We first computed the Mean Squared Error (MSE) for four models: a naive baseline (which predicts the next value as the previous one), a simple feedforward neural network, DeepAR, and the Temporal Fusion Transformer (TFT). The differences in MSE across models are relatively minor. Even the naive model performs comparably to more sophisticated ones, reinforcing the idea that additional architectural complexity does not necessarily translate into meaningful performance gains.
>
> To better understand this plateau, we estimated the learning curve $\Lambda(k)$. This curve is approximately flat and close to zero (see table). While our estimator is admittedly noisy and unstable, the results (shown below) suggest that there is little left to learn in this dataset.
>
> We then computed the minimal achievable risk using:
> $\hat{\mathcal{R}}^{\infty}(Q^{\*}) = \min_{1 \leq k \leq M} \hat{\mathcal{R}}^{k}(Q_k) - \hat{\Lambda}(k) $.
>
> Interestingly, the actual MSE values lie close to this estimated bound, suggesting that the performance ceiling is due less to model limitations than to the intrinsic unpredictability of the data.This aligns with the characteristics of the dataset. Unlike many other financial assets, FX rates are driven by the relative economic conditions, monetary policies, and geopolitical factors of two countries, adding significant complexity to their dynamics.
>
> **Table: Comparison between actual MSE and estimated minimal achievable risk for each model**
>
> | Model| MSE |$\hat{R}^{\infty}(Q^*) $|
> |---------------------|------------------------|----------------------------------------------|
> | Naive  | 0.000030 | 0.000021 ± 0.000007 |
> | DeepAR | 0.000018 ± 0.000004| 0.000013 ± 0.000009|
> | Simple FeedForward  | 0.000021 ± 0.000002    | 0.000015 ± 0.000008|
> | TFT | 0.000016 ± 0.000002| 0.000013 ± 0.000011|
> ---
>
> **W2 – Summary of Contributions**
>
> This work addresses two fundamental questions in sequential modeling:
>
> 1. **What is the minimal achievable risk when modeling sequential data?**
> 2. **Is poor predictive performance due to model limitations or to the intrinsic unpredictability of the data?**
>
> To address these questions, we introduce a practical framework grounded in information theory. A central element of this framework is the learning curve $\Lambda(k)$, which captures the improvement in predictive accuracy when the context length increases from $k$ to $k+1$. This measure is derived from the predictive information $I_{\text{pred}}(k, k')$ (defined as the mutual information between a block of $k$ past observations and a block of $k'$ future observations ) through this formula :
>
> $\Lambda(k) = \lim_{k' \to \infty} \left[ I_{\text{pred}}(k+1, k') - I_{\text{pred}}(k, k') \right]$
>
> Using this quantity, we introduce an estimator of  the *minimal achievable risk* — the lowest possible prediction error attainable by any model — through a bound involving the $k$-order forecasting risk. This estimator is defined as follows:
>
> $\hat{\mathcal{R}}^{\infty}(Q^{*}) = \min_{1 \leq k \leq M} \hat{\mathcal{R}}^{k}(Q_k) - \hat{\Lambda}(k)$,
>
> where $\hat{\mathcal{R}}^k(Q)$ is the empirical risk of a model $Q$ using a context of length $k$.
>
> This leads to a simple yet effective diagnostic tool to address the second question: is poor performance due to model limitations or to the data itself? If the actual model risk $\hat{\mathcal{R}}^k(Q)$ is close to the estimated minimal risk $\hat{\mathcal{R}}^\infty(Q^*)$, it indicates that either the data contains little predictive structure or that this structure has already been captured—suggesting that increasing model complexity is unlikely to yield better results. Conversely, a substantial gap between these two quantities signals that the model could be missing exploitable patterns in the data, pointing to potential model improvements.
>
> ---
> **Q1 - Corollary 4.4 Illustration**
>
> Due to recent NeurIPS policies, we are unfortunately unable to include new plots during the rebuttal period. However, we performed additional experiments to validate Corollary 4.4 using a concrete example that illustrates the asymptotic consistency of our dimension estimator.
>
> For this additional illustration, we follow the same setup as the experiment in **Section 5.3**, which is based on the Ising spin sequence example from [2]. We generated a sequence of $10^9$ samples with block size $M = 400{,}000$, and computed the learning curve up to $k = 20$. We restricted the analysis to $k \leq 20$ since $2^{20} \approx 10^6$ corresponds to the number of distinct words of length 20; beyond this point, entropy estimation $S(20)$ - via empirical frequencies - becomes unreliable due to insufficient sampling.
>
> The table below reports the values of our dimension estimator defined as $\hat{p}(k) = 2k \cdot \hat{\Lambda}(k)$ :
>
> | $k$|1|2|3|4|5|6|7|8|9|10|11|12|13|14|15|16|17|18|19|20|
> |---|---|---|---|---|---|---|---|---|---|---|---|---|---|---|---|---|---|---|---|---|
> | $\hat{p}(k)$|0.647|0.840|0.894|0.907|0.992|1.048|1.061|1.063|1.073|1.083|1.084|1.084|1.082|1.079|1.075|1.069|1.062|1.053|1.039|1.016|
>
> This clearly shows that our estimator converges to the real dimension p = 1.
>
> [1] provides both theoretical and empirical support for this behavior. In particular, they show that $\hat{p}(n) = 2n \cdot \Lambda(n) = p - \frac{1}{2n} + o\left(\frac{1}{n}\right)$, confirming that our estimator is consistent and can be used to estimate dimension of parameters, with convergence rate $(1/n)$.
>
> ---
>
> **Q2 - Estimating the true window size (model order) from small windows**
>
> Let us first restate the question to ensure we have understood it correctly.
>
> Given an autoregressive (AR) process of order $p$, **Proposition 4.2** in our paper shows that the learning curve $\Lambda(k)$ becomes exactly zero for all $k \geq p$. The reviewer asks whether it is possible to estimate this value of $p$ without having to compute $\Lambda(k)$ for all values up to $p$—in particular, using only small window sizes $k \ll p$.
>
> We propose a simple extrapolation-based method to approximate $p$ from a small number of evaluations. The learning curve $\Lambda(k)$ is decreasing and converges to zero, and under mild conditions—such as for Gaussian AR processes—it is also convex. We exploit this structure by computing $\Lambda(k)$ for $k = 1, \dots, M$, where $M$ is a user-defined limit, and fitting a simple parametric function such as a rational function $f(k) = a / (k + b)$ or an exponential decay. These function classes are chosen to reflect the smooth and typically convex decay of $\Lambda(k)$ as stated in [2]. Once the function is fitted, we define an estimator $\hat{p}$ as the first $k$for which $f(k) < \Lambda(1)/N$, with $N$ another tunable parameter controlling tolerance.
>
> To demonstrate this approach, we extended the experiment from **Section 5.2** of our paper. We generate a 5-dimensional Gaussian AR process with autoregressive coefficient $\rho = 0.9 $, and vary the true order $p $. We evaluate $\Lambda(k)$ for $k = 1$ to $10 $, fit the rational function $f(k)$, and estimate $\hat{p}$ as above with $M=10$ and $N = 10$. Results are summarized below:
>
> |$p$|10|50|100|500|1000|
> |---|---|---|---|---|---|
> |$\hat{p}$|10±0|51±3|98±5|459±18|811±49|
>
> This method achieves surprisingly accurate estimates, even for large values of $p$, despite using only the first 10 points of the learning curve. Of course, extrapolation becomes harder as $p$ grows, but the approach remains effective and tunable: one can adjust the number of observed points $M$, the threshold $N$, or the fitting function class to balance cost and precision.
>
> As an alternative, one could also apply a root-finding method (e.g., Newton’s method) to estimate where $\Lambda(k) \approx 0$. While such techniques may reduce the number of required evaluations, they still require accessing large $k$-values close to $p$, limiting their practical gain in high-order regimes.
>
> We thank the reviewer for this insightful question. These techniques may also prove useful in real-world applications where the learning curve is currently estimated via predictive information. Leveraging extrapolation strategies like the one discussed here could help refine learning curve estimation directly, beyond mutual information bounds.
>
>  ---
>
> **References**
>
> [1] Alexandrov, A., et al. (2019). *GluonTS: Probabilistic Time Series Modeling in Python*. arXiv preprint arXiv:1906.05264.
>
> [2] Bialek, W., Nemenman, I., & Tishby, N. (2001). *Predictability, complexity, and learning*.

---

### Official Review · Reviewer_PCUk · 2025-07-02

**Clarity:** 4
**Significance:** 3
**Originality:** 3
**Rating:** 4
**Confidence:** 3

**Summary:**

The paper introduces predictive information \( I_{\text{pred}} \), which measures the mutual information between a length-\(k\) past and a length-\(k'\) future in sequential data. It generalizes the EvoRate metric from \(k'=1\) to arbitrary future block lengths. The authors connect \( I_{\text{pred}} \),  with the concept of a universal learning curve \( \Lambda(k) \), which quantifies the incremental benefit of extending the past window by one step. Theoretically, this allows identifying the optimal context length and estimating the parameter dimension \(p\) of the underlying generative process. The method is evaluated on synthetic data, including Gaussian, autoregressive, and Ising processes, to validate its theoretical and empirical utility.

**Questions:**

Questions:

1. The Gaussian-process study reports results only for the fixed pair \((k = 5,\; k' = 10)\).  How was this setting chosen, and do your conclusions hold for other \((k,k')\) combinations?

2. What does the critical zone stand for in the learning-curve plot (Figure 2)?

3. Could you provide the correct reference for the definition of consistency in line 569?

**Ethical Concerns:**

["NO or VERY MINOR ethics concerns only"]

**Final Justification:**

After reading the authors’ detailed rebuttal, I believe they have sufficiently addressed all of my concerns. In particular they extended their experiments by showing results on a real world dataset, adding experimental details and explaining unclear parts of the work. If the commitee sees this work as a sufficiently large increment, I would definitely recommend accepting the paper.

**Limitations:**

yes

**Paper Formatting Concerns:**

-

**Quality:**

3

**Strengths And Weaknesses:**

Strengths:
The paper is very well written and presents a clear theoretical contribution. It offers exhaustive derivations and proofs, and it generalizes EvoRate while drawing a formal connection to information theory through the universal learning curve.

Weaknesses:
- Limited empirical scope: currently all experiments are performed on synthetic data. No real-world sequential datasets are used to show practical benefit. This makes it hard for me to assess the impact of this work apart from the theoretical work and the mapping of predictive information and information theory. However, it is to note that the authors mention the lack of real-world data experiments as a limitation in the corresponding section.

- Expressiveness of the experiments: Although the chosen synthetic experiments are well designed, it is unclear whether the reported benefits would persist under a wider set of settings, for example, more \((k,k')\) combinations in the first experiment, additional \(p\) values in the second, or cross-validation splits in the third.

- Reproducibility: for the last experiment using MLP and LSTM the exact model parameter counts are not provided. Including these numbers would make the results easier to reproduce and interpret.

---

> ### Author Rebuttal · Authors · 2025-07-28
>
> We thank the reviewer for their overall positive evaluation and for highlighting both the strengths and limitations of our work. We appreciate the comments regarding empirical scope, experimental coverage, and reproducibility, and we address each of these points below.
>
> **W1 – Limited empirical scope**
>
> It is true that our paper does not include any experiments on real-world datasets and we understand that it might be challenging to fully assess the impact of our work without those. The primary focus of our work is indeed theoretical, and as you may have seen, the theoretical contributions—particularly the proofs—are already substantial and technically demanding.
>
> We intentionally chose not to simultaneously tackle the separate and equally challenging problem of experimentally estimating the learning curve and the minimal achievable risk. Indeed, estimating the learning curve is itself a complex theoretical question that warrants dedicated treatment. Addressing it rigorously would require an entirely separate line of work, and we felt that attempting to incorporate this into the current paper would dilute the clarity and focus of our main contributions. For instance, [1] is a full research article dedicated solely to estimating mutual information in sequential settings. Hence, we chose to first address the problem from a theoretical perspective, and plan to tackle the estimation of the learning curve and thus minimal risk in a follow-up article.
>
> That said, we did include experiments on synthetic data, where the estimation of $\mathbf{I}_{\text{pred}}$ is not an issue, since closed-form expressions are available. These experiments serve an illustrative purpose, allowing us to demonstrate the theoretical framework in controlled settings.
>
> Nevertheless, we fully acknowledge the reviewers' concern regarding the lack of real-world evaluations. In response, we applied our framework to the Exchange Rate dataset from GluonTS [2], and show how it can be used on a real world dataset. This dataset contains daily exchange rates (1990–2016) for eight major currencies (AUD, GBP, CAD, CHF, CNY, JPY, NZD, SGD). In this case, we show that this dataset is more limited by data itself rather than by model complexity.
>
> We first computed the MSE for four models: a naive baseline (which predicts the next value as the previous one), a simple feedforward neural network, DeepAR, and the Temporal Fusion Transformer (TFT). The differences in MSE across models are relatively minor. Even the naive model performs comparably to more sophisticated ones, reinforcing the idea that additional architectural complexity does not necessarily translate into meaningful performance gains.
>
> To obtain the  minimal achievable risk $\hat{\mathcal{R}}^{\infty}(Q^{*}) = \min_{1 \leq k \leq M} \hat{\mathcal{R}}^{k}(Q_k) - \hat{\Lambda}(k)$, we estimated the learning curve $\Lambda(k)$.  While our estimator is admittedly noisy and unstable, the results (shown below) suggest that there is little left to learn in this dataset. Indeed, the actual MSE values lie close to this estimated minimal risk, suggesting that the performance ceiling is due less to model limitations than to the intrinsic unpredictability of the data.
>
> This aligns with the characteristics of the dataset. Unlike many other financial assets, FX rates are driven by the relative economic conditions, monetary policies, and geopolitical factors of two countries, adding significant complexity to their dynamics. This market is also extremely liquid and highly competitive, so any patterns or inefficiencies are quickly detected and eliminated by market participants. These factors contribute to the inherent unpredictability of the dataset.
>
> **Table: Comparison between actual MSE and estimated minimal achievable risk for each model**
>
> | Model| MSE |$\hat{R}^{\infty}(Q^*) $|
> |---------------------|------------------------|----------------------------------------------|
> | Naive  | 0.000030 | 0.000021 ± 0.000007 |
> | DeepAR | 0.000018 ± 0.000004| 0.000013 ± 0.000009|
> | Simple FeedForward  | 0.000021 ± 0.000002    | 0.000015 ± 0.000008|
> | TFT | 0.000016 ± 0.000002| 0.000013 ± 0.000011|
>
> ---
>
> **W2 – Expressiveness of the experiments**
>
> For the $(k, k')$ combinations, please refer to our response to **Question 1**.
> Regarding the second experiment, the results hold consistently across all values of $p$. We chose to report only $p = 5$ and $p = 10$ in the article to maintain clarity and avoid cluttering the figure.
> For the third experiment, we already included several cross-validation block sizes ( $\(M = 10^4, 10^5, 10^6, 10^7\)$ ) to illustrate the robustness of the results across different magnitudes.  That said, our conclusions remain valid for other values of $M$ as well.  We do observe, however, that the estimation of the learning curve $\Lambda(k)$ becomes less stable when $M \ll 10^4$, due to higher variance in the empirical estimator of $\mathbf{I}_{\text{pred}}$
>
> ---
>
> **W3 – Reproducibility (for the last experiment)**
>
> Here are the exact model and training details:
>
> **Model architectures**
> - **MLP**: 2 hidden layers (64, then 32 neurons), ReLU activations, output layer with 2 units (softmax).
> - **LSTM**: Single-layer LSTM (32 hidden units), fully connected layer mapping to 2 output classes.
>
> **Training setup**
> - Adam optimizer (lr = $10^{-3}$)
> - Batch size = 128
> - Max epochs = 1000
> - Early stopping with patience = 10
> - 50 batches per epoch
> - 80/20 train-validation split
>
> We will explicitly add these details in the final manuscript.
>
> ---
> **Q1 – On the fixed pair $(k, k')$**
>
> We selected the fixed pair $(k, k')$ = $(5, 10)$ based on two main considerations.
>
> First, we needed to satisfy the constraint $k'$ > $k$, and since the rest of the paper focuses on estimating the learning curve $\Lambda(k)$  we aimed for $k'$ to be sufficiently large to approximate the limit $k' \to \infty$.
> Second, for autoregressive Gaussian processes with order $1$ < d < $20$, we wanted $k$ and $k'$ to be of the same order of magnitude as $d$, which motivated our choice of $k$ = $5$ and $k'$ = $10$. While this choice is somewhat arbitrary, other settings such as $(k, k')$ = $(10, 20)$ would also be valid.
>
> Importantly, our conclusions hold for other combinations. We repeated the first experiment with other $(k, k')$ pairs and observed consistent results. Due to character limits, we report only the results for $(k, k') = (10, 20)$ here, but we can provide additional tables upon request. We had to remove some columns from the original experiment due to space and formatting constraints.
>
> ---
>
> **Table: Estimated predictive information $ \hat{I}_{\text{pred}}$ across models and GP kernels for $(k, k') = (10,20)$**
> | |AR (d=1)|AR (d=10)|Gaussian (ρ=0.5, d=1)|Gaussian (ρ=0.5, d=10)|Gaussian (ρ=0.8, d=1)|Gaussian (ρ=0.8, d=10)|Local-P (d=1)|Matérn 3/2 (d=1)|Matérn 3/2 (d=10)|Matérn 5/2 (d=1)|Matérn 5/2 (d=10)|Periodic (d=1)|RBF (d=1)|RQ (d=10)|RQ (Extra)|
> |--|--------|---------|----------------------|-----------------------|----------------------|-----------------------|--------------|------------------|-------------------|------------------|-------------------|---------------|---------|----------|-----------|
> |$\hat{\mathbf{I}}_{\text{pred}}\text{-}\mathtt{DV}$|0.49|3.08|0.14|1.11|0.51|3.51|4.95|0.60|4.20|0.93|4.94|4.32|3.28|6.29|4.89|
> |$\hat{\mathbf{I}}_{\text{pred}}\text{-}\mathtt{NWJ}$|0.50|3.49|0.13|1.24|0.48|3.43|4.13|0.60|3.76|0.94|4.13|4.17|3.59|4.21|4.09|
> |$\hat{\mathbf{I}}_{\text{pred}}\text{-}\mathtt{SMILE}$|0.49|3.00|0.14|1.01|0.49|2.85|4.57|0.60|3.07|0.95|2.21|3.66|2.98|1.95|3.76|
> |$\hat{\mathbf{I}}_{\text{pred}}\text{-}\mathtt{InfoNCE}$|0.51|3.48|0.14|0.87|0.49|3.78|4.87|0.58|4.48|0.95|6.34|4.24|3.00|9.54|5.01|
> |$\hat{\mathbf{I}}_{\text{pred}}\text{-}\mathtt{TUBA}$|0.50|3.27|0.14|1.11|0.49|2.74|4.68|0.60|2.96|0.96|3.34|3.52|3.10|2.97|3.66|
> |$\hat{\mathbf{I}}_{\text{pred}}\text{-}\mathtt{TRUE}$|0.51|5.11|0.14|1.44|0.51|5.11|8.57|0.62|6.17|1.00|10.00|12.00|6.34|15.91|7.9|
>
> ---
>
> **Q2 – Meaning of the "critical zone" in Figure 2**
>
> The "critical zone" represents values of the universal learning curve $\Lambda(k)$ below a small threshold (here set at $0.02$).
>
> In practice, estimating $\Lambda(k)$ doesn't yield exact zeros due to inherent variance. Therefore, choosing precisely the context length $k$ at which the learning curve becomes negligible is challenging.
>
> To address this, we introduce a practical cutoff—this "critical zone"—below which we consider the learning curve effectively zero. In our example, the threshold ($0.02$) was chosen to be small relative to the typical magnitude of $\Lambda(k)$, yet large enough to account for estimation noise.
>
> While somewhat arbitrary, this choice helps clearly identify the region where additional context no longer meaningfully improves predictions.
>
> ---
>
> **Q3 – Reference correction (line 569)**
>
> Thank you for catching this error. The missing reference is [3].
> We will correct this citation in the final version.
>
> ---
>
> We thank the reviewer again for their constructive feedback. We believe the clarifications above, as well as the new real-world experiment on the Exchange Rate dataset, help to address the concerns raised regarding empirical scope and reproducibility.
>
> **References**
>
> [1] Zeng, Q., Huang, L.-K., Chen, Q., Ling, C. X., & Wang, B. (2025). *Towards Understanding Evolving Patterns in Sequential Data*. Advances in Neural Information Processing Systems.
>
> [2] Alexandrov, A., et al. (2019). *GluonTS: Probabilistic Time Series Modeling in Python*. arXiv preprint arXiv:1906.05264.
>
> [3] Kallenberg, O. (2002). *Foundations of Modern Probability*. Springer.

---

> > ### Comment · Reviewer_PCUk · 2025-08-05
> >
> > Thank you for your exhaustive response!
> >
> > W1:
> > Thanks for providing the additional experiment. I understand that the application on real world datasets is another challenging dimension and therefore is not in scope of this work. I suggest to include the experiment on the Exchange Rate dataset as a preliminary work that can be extended.
> >
> > Q1/W2:
> > Thank you for the answer!  The results for Q1 look convincing that the pattern applies to different settings of k and k’.
> > I would encourage you to provide those additional results, as well as the ones for different p, in your paper.
> >
> > W3:
> > Thanks for adding the additional information in your work!
> >
> > Q2:
> > Thank you for the explanation. I understand that the choice is somewhat arbitrary. Maybe this could be enhanced by a short mention in the text as to what the critical-zone refers to, as it is important decision where to set it.
> >
> > That being said, I am satisfied with your response!
> > However, I would like to note that I am not an expert in this specific field, which makes it hard for me to judge how incremental the contribution is and whether it is sufficiently large. I will therefore defer the final decision to the more senior members of the committee. From my perspective, this is a technically strong theoretical paper, supported by well-designed and convincing experiments.

---

> > > ### Author Response · Authors · 2025-08-05
> > >
> > > Thank you for taking time to read our rebuttal. We are glad that our responses helped clarify your concerns. We appreciate your suggestions and will incorporate the additional experiments and clarifications you recommended in the revised version of the paper.

---

### Official Review · Reviewer_MaER · 2025-07-02

**Clarity:** 3
**Significance:** 3
**Originality:** 2
**Rating:** 5
**Confidence:** 3

**Summary:**

Building on the classical works of Bialek and Tishby [1], and Crutchfield and Feldman [2], this paper addresses the following questions related to sequential modeling:

1)	What is the minimum achievable risk for a predictor modeling sequential data?
2)	If a model performs poorly, is it due limitations of the model, or due to the inherent unpredictability of the data?

Effective sequence modeling requires (1) predictable patterns in the data, and (2) a model capable of exploiting those patterns. Determining whether a sequence contains predictable patterns is a crucial first step. Predictive information $I_{\text{pred}}(k, k')$ (mutual information between the past $k$ observations and the future $k'$ observations) and universal learning curve $\Lambda(k)$ (the discrete derivative of predictive information as $k' \to \infty$) are classical quantities that measure the learnability and complexity of a sequence.

The paper begins by revisiting basic concepts from information theory and defining predictive information and the universal learning curve under the assumptions of stationarity and finite entropy. The authors then compute these quantities for a $m$-order Markov chain and a parametric process where the dimensionality of the parameter space is $p$. Similar calculations can also be found in [1, 2].

Next, the authors bound the minimal achievable risk by the true $k$-th order forecasting risk minus $\Lambda(k)$. They also present an empirical version of this bound involving the empirical $k$-th order forecasting risk. Based on this bound, if the model's loss fails to improve beyond a certain context length $k$ while $\Lambda(k) > 0$, it indicates that the current model complexity is insufficient to capture the learnable patterns in the sequence. This, I believe, is the key contribution of the paper, as it addresses questions (1) and (2) above.

Finally, the authors provide some experimental results demonstrating the estimation of $I_{\text{pred}}(k, k')$ on a Gaussian process, the ability of the estimated $\Lambda(k)$ to identify the correct order of an autoregressive process, and the estimation of the minimum achievable forecasting risk of an Ising spin sequence.

**References**

[1] William Bialek and Naftali Tishby. Predictive information. arXiv preprint cond-mat/9902341, 352 1999.

[2] James P Crutchfield and David P Feldman. Regularities unseen, randomness observed: Levels of entropy convergence. Chaos: An Interdisciplinary Journal of Nonlinear Science, 13(1): 25–54, 2003.

**Questions:**

I'm listing some minor questions and suggestions below.

1. In line 32, you cite an example about the performance discrepancy related to GluonTs despite recent innovations. Can you elaborate this example by proving more context?

2. In line 139, you use $\ln$ and later in Eq. 6, you use $\log$. Also, in line 139, $\mathcal{F}_k$ is a set of functions, and in line 186, $\mathcal{F}$ is the Fisher information matrix. Better to be consistent with the notation.

3. In Figure 1, you have not indicated what the strength of the color means.

4. Prospective Learning [1] introduces a framework in which data is generated by a stochastic process, and the learner is expected to produce a sequence of hypotheses that achieves low risk on future observations, given the data observed up to the present. I believe some of the results from your paper could be used to extend or enrich this framework. Related ideas have also been explored in [2, 3]. It would be valuable to position your work in the context of these studies in the related work section.

**References**

[1] De Silva, A., Ramesh, R., Yang, R., Yu, S., Vogelstein, J. T., & Chaudhari, P. (2024). Prospective Learning: Learning for a Dynamic Future. arXiv preprint arXiv:2411.00109.

[2] Dawid, A. P., & Tewari, A. (2020). On learnability under general stochastic processes. arXiv preprint arXiv:2005.07605.

[3] Hanneke, S. (2021). Learning whenever learning is possible: Universal learning under general stochastic processes. Journal of Machine Learning Research, 22(130), 1-116.

**Ethical Concerns:**

["NO or VERY MINOR ethics concerns only"]

**Final Justification:**

The authors have successfully addressed the concerns I had. This paper employs predictive information and universal learning curve initially introduced by Bialek and Tishby, and Crutchfield and Feldman, to quantify the limits of sequential learning. I will maintain my score in favor of accepting the paper.

**Limitations:**

yes.

**Paper Formatting Concerns:**

None.

**Quality:**

3

**Strengths And Weaknesses:**

I enjoyed reading this paper. The ideas are communicated clearly and effectively. However, I believe it lacks a bit of originality, as tools such as predictive information (or excess entropy) and the universal learning curve (or entropy gain) are well-established concepts in the field. That said, the paper makes a novel and interesting contribution by connecting these quantities to the minimum achievable risk, which adds valuable insight to sequential learning and modeling.

---

> ### Author Rebuttal · Authors · 2025-07-28
>
> We would like to thank the reviewer for the careful reading and constructive feedback. Below we address each question in turn.
>
> ---
>
> **Q1 – GluonTS**
>
> The example mentioned in line 32 refers to the Exchange Rate dataset from GluonTS [1], which contains daily exchange rates (1990–2016) for eight major currencies (AUD, GBP, CAD, CHF, CNY, JPY, NZD, SGD).
>
> In this dataset, we observe that even state-of-the-art models — including Transformer-based and probabilistic architectures — offer only marginal improvements over simpler baselines such as DeepAR. This suggests a plateau in forecasting performance despite increasing model complexity.
>
> To support this observation, we computed Exchange dataset MSE for four models: a naive baseline (which predicts the next value as the previous one), a simple feedforward neural network, DeepAR, and the Temporal Fusion Transformer (TFT) and applied our framework to this dataset. As shown in the table below, the differences in MSE across models are relatively minor. Even the naive model performs comparably to more sophisticated ones, reinforcing the idea that additional architectural complexity does not necessarily translate into meaningful performance gains.
>
> To obtain the  minimal achievable risk $\hat{\mathcal{R}}^{\infty}(Q^{*}) = \min_{1 \leq k \leq M} \hat{\mathcal{R}}^{k}(Q_k) - \hat{\Lambda}(k)$, we estimated the learning curve $\Lambda(k)$.  While our estimator is admittedly noisy and unstable, the results (shown below) suggest that there is little left to learn in this dataset. Indeed, the actual MSE values lie close to this estimated minimal risk, suggesting that the performance ceiling is due less to model limitations than to the intrinsic unpredictability of the data.
>
> This aligns with the characteristics of the dataset. Unlike many other financial assets, FX rates are driven by the relative economic conditions, monetary policies, and geopolitical factors of two countries, adding significant complexity to their dynamics. This market is also extremely liquid and highly competitive, so any patterns or inefficiencies are quickly detected and eliminated by market participants. These factors contribute to the inherent unpredictability of the dataset.
>
> **Table: Comparison between actual MSE and estimated minimal achievable risk for each model**
>
> | Model| MSE |$\hat{R}^{\infty}(Q^*) $|
> |---------------------|------------------------|----------------------------------------------|
> | Naive  | 0.000030 | 0.000021 ± 0.000007 |
> | DeepAR | 0.000018 ± 0.000004| 0.000013 ± 0.000009|
> | Simple FeedForward  | 0.000021 ± 0.000002    | 0.000015 ± 0.000008|
> | TFT | 0.000016 ± 0.000002| 0.000013 ± 0.000011|
>
> ---
> **Q2 – Notations**
>
> Thank you for pointing this out. We will revise the final version to avoid overloading notation (e.g., $\mathcal{F}_k$ vs. $F$) and clarify the use of logarithms throughout.
>
> ---
>
> **Q3 – Figure 1 legend**
>
> In the final version, we will clarify that blue regions indicate negative bias (underestimation), and that color intensity reflects the magnitude of this bias.
>
> ---
>
> **Q4 – Related work section**
>
> We are grateful to the reviewer for bringing up the notion of prospective learning, which was not initially part of our analysis, and for pointing us to relevant references. We will incorporate the following discussion into the "Related Work" section:
>
> Recent developments in statistical learning theory increasingly emphasize learning in contexts where classical assumptions, such as independent and identically distributed (i.i.d.) samples, no longer hold. In particular, extending traditional frameworks to accommodate stochastic processes with temporal dependencies or evolving distributions has become a critical direction for research.
> [2] directly addresses this by proposing learners that output sequences of hypotheses aimed at minimizing future risk—an approach well-suited for non-stationary settings where the optimal predictor may itself change over time. Complementary to this, [3, 4] offer general characterizations of learnability under arbitrary stochastic processes, framing it in terms of regret minimization or universal consistency, and showing its deep connections with online learning.
> These frameworks share a common goal: to define when learning is possible in complex, non-i.i.d. environments. However, they primarily rely on external performance criteria—such as risk or regret—without explicitly analyzing the structural properties of the data itself. Our work complements these approaches by introducing an intrinsic, information-theoretic perspective: we characterize learnability in terms of the presence of stable and identifiable patterns within the data-generating process. This allows us to bridge high-level learnability guarantees with the underlying structure of the data, providing new insights into how and why generalization is achievable beyond traditional settings.
>
> ---
> We hope the clarifications above resolve the reviewer’s concerns and further illustrate how our framework complements existing work on sequential learning.  We appreciate the insightful feedback and look forward to refining the final manuscript accordingly.
>
>
> **References**
>
> [1] Alexandrov, A., et al. (2019). *GluonTS: Probabilistic Time Series Modeling in Python*. arXiv preprint arXiv:1906.05264.
>
> [2] De Silva, A., Ramesh, R., Yang, R., Yu, S., Vogelstein, J. T., & Chaudhari, P. (2024). *Prospective Learning: Learning for a Dynamic Future*. arXiv preprint arXiv:2411.00109.
>
> [3] Dawid, A. P., & Tewari, A. (2020). *On learnability under general stochastic processes*. arXiv preprint arXiv:2005.07605.
>
> [4] Hanneke, S. (2021). *Learning whenever learning is possible: Universal learning under general stochastic processes.* Journal of Machine Learning Research, 22(130), 1-116.

---

> > ### Comment · Reviewer_MaER · 2025-08-07
> > **Response to the Rebuttal**
> >
> > Dear authors,
> >
> > Thank you for explaining the context behind GluonTS observation and addressing the other concerns I had. I would further suggest that you make it clear in the beginning of the paper that quantities such as predictive information and universal learning curve are not novel and that they have been introduced and discussed in detail in the works of of Bialek and Tishby [1], and Crutchfield and Feldman [2]. I will maintain my score.
> >
> > [1] William Bialek and Naftali Tishby. Predictive information. arXiv preprint cond-mat/9902341, 352 1999.
> >
> > [2] James P Crutchfield and David P Feldman. Regularities unseen, randomness observed: Levels of entropy convergence. Chaos: An Interdisciplinary Journal of Nonlinear Science, 13(1): 25–54, 2003.

---

> > > ### Author Response · Authors · 2025-08-08
> > >
> > > Dear reviewer,
> > >
> > > Thank you for taking time to read our rebuttal and for the suggestions you gave. We will take them in account for the final manuscript.

---

### Decision · Program_Chairs · 2025-09-17

**Decision:**

Accept (poster)

**Comment:**

This paper proposes an interesting predictive information framework for quantifying the learnability in sequential data. Specifically,  the authors  formalize this by using the predictive mutual information between the past $k$ and the future $k'$ observations, as well as the   connection with the universal learning curve Λ(k). The theoretical contributions based on the empirical risk are interesting,  while synthetic datasets, e.g. in Gaussian processes, show the ability to detect optimal autoregressive orders.

All reviewers agree that this is a strong submission that provides an rigorous information theoretic method to quantity
learnability in sequential data settings and do Markov order selection.